# Impact of Infection with *Flavobacterium psychrophilum* and Antimicrobial Treatment on the Intestinal Microbiota of Rainbow Trout

**DOI:** 10.3390/pathogens12030454

**Published:** 2023-03-14

**Authors:** David Huyben, Maureen Jarau, Janet MacInnes, Roselynn Stevenson, John Lumsden

**Affiliations:** 1Department of Animal Biosciences, University of Guelph, Guelph, ON N1G 2W1, Canada; 2Department of Pathobiology, University of Guelph, ON N1G 2W1, Canada; 3Department of Molecular and Cellular Biology, University of Guelph, Guelph, ON N1G 2W1, Canada

**Keywords:** *Flavobacterium psychrophilum*, rainbow trout, florfenicol, erythromycin, experimental infection, intestinal microbiota

## Abstract

The diversity and composition of intestinal microbiota in rainbow trout have been studied using next-generation sequencing (NGS), although few studies have examined the effects of antimicrobials. We evaluated the effect of antibiotics florfenicol and erythromycin and infection with or without *Flavobacterium psychrophilum* on the intestinal microbiota in rainbow trout juveniles (30–40 g) using NGS. Prophylactic oral antibiotic treatments were administered for 10 days before groups of fish were injected intraperitoneally with virulent *F. psychrophilum*. Intestinal content (allochthonous bacteria) was collected at day −11, 0, 12, and 24 p.i., and the v3–v4 region of the 16S rRNA gene was sequenced using Illumina MiSeq. Before prophylactic treatment, Tenericutes and Proteobacteria were the most abundant phyla identified and *Mycoplasma* was the most abundant genus. Fish infected with *F. psychrophilum* had decreased alpha diversity and a high abundance of *Mycoplasma*. Fish administered florfenicol had increased alpha diversity compared to the control at day 24 p.i., although both florfenicol and erythromycin-treated fish had a higher abundance of potential pathogens, specifically *Aeromonas*, *Pseudomonas,* and *Acinetobacter*. Mycoplasma disappeared after treatment but appeared again after day 24. This study demonstrates that prophylactic oral treatment with antibiotics florfenicol and erythromycin as well as *F. psychrophilum* infection changed the composition of intestinal microbiota in rainbow trout juveniles that did not recover by day 24 p.i. and further long-term effects on the host need to be investigated.

## 1. Introduction

The microbes in the gastrointestinal tract, termed intestinal microbiota, play important roles in the metabolism, nutrition, and immunity of animals, including fish. Early studies of fish gut microbiota used culture-dependent methods [1,2], which were limited since not all microbes can be cultured, and those that can are often overgrown by fast-growing bacteria. In recent years, most microbiota studies have been performed using culture-independent methods, e.g., next-generation sequencing (NGS) technologies on Illumina platforms [3].

Rainbow trout (*Oncorhynchus mykiss*) is one of the most popular freshwater finfish, both farmed and pursued in the sport fishing industry worldwide [4]. In the past decade, the intestinal microbiota of rainbow trout has been characterized using NGS. At the phyla level, bacteria are commonly represented by Proteobacteria, Firmicutes, Tenericutes, and Bacteroidetes while at the genus level *Lactobacillus*, *Streptococcus*, *Bacillus*, *Mycoplasma*, *Vibrio*, *Pseudomonas* and *Photobacterium* are common [5,6,7,8,9]. Several studies have identified the bacteria present in the rainbow trout intestine and the influence of diet using NGS, although research is lacking on the effects of disease and antibiotic treatment, especially over time.

Rainbow trout are highly susceptible to rainbow trout fry syndrome (RTFS) and bacterial cold-water disease (BCWD) later on as juveniles and adults. These diseases are caused by the Gram-negative filamentous bacterium *Flavobacterium psychrophilum* predominantly between 6–10 °C. RTFS and BCWD have been associated with high economic and production losses in the aquaculture industry since it has been reported to cause up to 90% mortality in fry and up to 20% mortality in juvenile and adult fish [10,11]. Antimicrobial therapy with florfenicol is commonly used to control and treat RTFS and BCWD outbreaks; however, its’ application not only promotes the transmission of antimicrobial-resistant bacteria and genes but also affects the balance of beneficial microbes that reside in the gastrointestinal tract [12]. Antimicrobials can also deregulate metabolism, compromise immune homeostasis, and increase susceptibility to infection [12]. Recently, a few studies have highlighted the impact of antibiotics, such as oxytetracycline and enrofloxacin, on the communities of intestinal microbiota in rainbow trout using NGS [13,14,15] and infection with *F. psychrophilum* [16]. However, only one study has investigated the effects of florfenicol and *F. psychrophilum* infection over time on rainbow trout fry [17]. No studies have investigated the effects of erythromycin, which has demonstrated efficacy for experimental BCWD [18] but is only available on an emergency basis for fish in Canada.

Understanding the impact of antimicrobial agents on the intestinal microbiota composition of rainbow trout could aid in improving farm management and lead to reduced antimicrobial use and thus enhance the welfare and safety of rainbow trout for human consumption. Florfenicol is often used to treat BCWD as soon as feed intake drops for a sustained period as waiting longer in an outbreak often leads to high mortality and multiple treatments. Strictly speaking, this would be a prophylactic treatment and we were interested in the impact on a subsequent infection that this use would have. Thus, the objectives of this study were (i) to evaluate the effect of prophylactic oral florfenicol (FFN) and erythromycin (ERY) treatment on the composition and diversity of intestinal microbiota in rainbow trout juveniles, and (ii) to investigate the susceptibility of prophylactically treated fish to *F. psychrophilum* infection.

## 2. Material and Methods

### 2.1. Fish and Rearing Conditions

Rainbow trout juveniles weighing approximately 30–40 g were obtained from Lyndon Fish Hatcheries Inc. (New Dundee, ON, Canada) and housed in the Hagen Aqualab facility, University of Guelph, ON, Canada. Fifty fish were kept in each of the 12 circular fiberglass tanks (125 L) with continuous aeration and flow-through groundwater at a rate of 2 L min^−1^ with an adjusted temperature of 10.0 ± 1.0 °C. The photoperiod was 12 h dark:light and fish were fed once daily by hand with a commercial trout feed pellet (Profishent, Martin Mills Inc., Elmira, ON, Canada) at a rate of 1.5% of biomass day^−1^ for the duration of the trial. Daily observations were conducted to evaluate fish behavior as well as morbidity and mortality. Prior to the start of the experiment, the fish were acclimated for at least one week in the biocontainment room. Once acclimated, five fish were euthanized using benzocaine solution (ethyl-4-aminobenzoate; 50 mg L^−1^) for general health screening by gross examination and tested for the presence of *F. psychrophilum* in spleen and kidney by bacterial culture. Identification of isolates was confirmed using a 16S rRNA gene qPCR assay [19]. All animal procedures and protocols were approved by the University of Guelph Animal Care Committee and were in accordance with the guidelines of the Canadian Council of Animal Care.

### 2.2. Preparation of Medicated Feed

Medicated fish feed was prepared according to Jarau, et al. [20]. The commercial powdered antibiotic preparations Aquaflor^®^ (Merck, Rahway, Lavaltrie, NJ, USA) and Gallimycin^®^-50 (Vetoquinol N.A Inc, QC, Canada) were used to prepare the FFN- and ERY-medicated diets, respectively. The medicated diet for each treatment group was prepared based on the treatment dose of each antibiotic. FFN was administrated at 10 mg kg^−1^ fish body weight (BW) day^−1^ and ERY at 75 mg kg^−1^ fish BW day^−1^. The medicated feeds were prepared by surface coating the antibiotics with oil on the commercial fish feed pellet (Profishent 3.0 mm) following the manufacturer’s instructions. Once dried, the coated feed was dispensed into small bags (1 bag tank^−1^) and stored at −20 °C until use. Control feed was surface coated with oil without any antibiotics.

### 2.3. Preparation of Bacterial Inoculum

A virulent strain of *F. psychrophilum*, FPG101 [20], was isolated in 2008 from the kidney of a rainbow trout with BCWD and used in this study. The bacterial maintenance and preparation of the *F. psychrophilum* inoculum were performed as described by Jarau, Di Natale, Huber, MacInnes, and Lumsden [20]. For the preparation of experimental infection material, frozen stocks were first streaked on Cytophaga agar (CA) plates and incubated at 15 °C for 96 h to obtain single colonies. Five single colonies were then subcultured on CA plates and incubated at 15 °C for 72 h and cells were harvested and suspended in CB. The cell suspensions were adjusted using a Novaspec Plus spectrophotometer (GE Healthcare Life Sciences, Mississauga, ON, Canada) to an optical density of 0.6 at 600 nm which corresponded to ~2.24 × 10^9^ colony forming units (cfu) mL^−1^. Then, the suspension was diluted to obtain a concentration of ~1 × 10^9^ cfu mL^−1^. To confirm the cell numbers, suspensions were enumerated by plating 0.1 mL of 10-fold serial dilutions on CA plates (in triplicates per dilution). After incubation at 15 °C for 4 d, visible colonies were counted on replicate plates containing 25–250 colonies and the number of cfu mL^−1^ was calculated [21].

### 2.4. Experimental Design

The details of fish groups and the sampling timeline used in this study are shown in Table 1. The experimental trial involved two treatment periods; starting with the prophylactic oral treatment period (10 days of oral antibiotic treatment) followed by the infection period. Fish were divided into four groups and each group was placed in triplicate tanks with 50 fish in each tank. During the 10 days prophylactic oral treatment period, there were two groups that were treated with either FFN (group F) or ERY (group E), and the remaining groups were not treated with any antimicrobials (Ca & Cb). Following the treatment period, three groups [two treated groups (F & E) and an untreated group (Cb)] were injected with *F. psychrophilum* FPG101 whereas untreated group Ca was mock-infected with CB and served as control. Due to tank space limitations, we did not include a control group of fish that were not infected and administered antibiotics.

### 2.5. Prophylactic Oral Antibiotic Treatment

The two treated fish groups were fed medicated feed prepared as described by Jarau, Di Natale, Huber, MacInnes, and Lumsden [20]. Medicated feed was given at a rate of 1.5% of fish body weight per d for 10 consecutive days. Non-treated and control fish groups were fed with control feed at the same rate. The response to feeding was recorded and uneaten feed was removed after 15 min.

### 2.6. Flavobacterium Psychrophilum Infection

Prior to infection, fish were anesthetized with benzocaine (50 mg L^−1^). Infected groups were injected intraperitoneally with *F. psychrophilum* FPG101 at a dose of 1 × 10^8^ cfu fish^−1^ and a control group were mock-infected with 0.1 mL of sterile CB. This dose was found previously to induce 51% mortality in rainbow trout (Jarau et al., 2018 [20]). Following infection, fish were observed 3x d^−1^ for any signs of morbidity and mortality. Dead and moribund fish with signs characteristic of BCWD infection was removed from the tank, humanely euthanized, and recorded as mortalities. Spleen samples were cultured on CB to determine the presence of *F. psychrophilum.* The identity of putative *F. psychrophilum* isolates was confirmed using an *F. psychrophilum*-specific qPCR assay targeting the 16S rRNA gene [19].

### 2.7. Sample Collection

The intestinal contents from 3 fish tank^−1^ (n = 9 treatment^−1^) were sampled at four-time points; -D11, D0, D12 & D24 post-infection (p.i.) as outlined in Figure 1. Spleen samples from 3 fish tank^−1^ (n = 9 treatment^−1^) were collected on D3, D6, D9 and D12 p.i. One day prior to sampling, all fish groups fasted. Sampled fish were euthanized using benzocaine and moved to the laboratory for dissection. Prior to dissection, the ventral surface of the fish was sprayed with 70% ethanol. Intestinal contents (allochthonous bacteria) and spleen collection were performed as described below.

#### 2.7.1. Intestinal Contents

For the microbiota analysis, intestinal contents were collected from individual fish. The abdominal body wall and the tissues surrounding the digestive tract were removed aseptically. The intestinal contents were carefully removed by squeezing the intestine with sterile forceps into a sterile 1.5 mL microcentrifuge tube. The tubes were snap frozen in liquid nitrogen and immediately stored at −80 °C until use.

#### 2.7.2. Spleen

To determine the bacterial load at the selected time points, spleens were collected on p.i. D3, D6, D9 and D12. These were not the same fish sampled for intestinal content and a maximum of 12 days was used based on previous detectable levels of *F. psychrophilum* by qPCR [20]. The load of *F. psychrophilum* was quantified using a *rpoC* qPCR assay [22]. Spleens were aseptically removed and placed in sterile 1.5 mL microcentrifuge tubes containing RNAlater. The tubes were then incubated for 24 h at room temperature and stored at −80 °C until used.

### 2.8. Microbiota Analysis

#### 2.8.1. Intestinal Contents DNA Extraction

Total genomic DNA from intestinal contents was extracted using the QIAmp DNA Stool Mini Kit (Qiagen, Toronto, ON, Canada) following the manufacturer’s spin column protocol for pathogen detection. DNA was extracted from ~180 to 220 mg of fecal matter. The protocol included lysis of the sample using buffer ASL, vortexed, heat-lysis at 95 °C for 5 min, and treatment of impurities using the InhibitEx matrix before purification of the DNA by spin column. The purity and concentration of the extracted DNA were determined using a Nanodrop^™^ 2000 spectrophotometer (Thermo Fisher Scientific, Mississauga, ON, Canada). The extracted DNA was stored at −20 °C until use. A negative control sample was not included in DNA extraction, PCR amplification, or sequencing and should be included in future studies to assure no contamination.

#### 2.8.2. 16S rRNA Gene PCR Amplification and Purification

The 16S rRNA gene sequencing library was prepared following the “16S Metagenomics Sequencing Library Preparation” protocol (Illumina, San Diego, CA, USA) targeting the V3 to V4 region of the 16S rRNA gene as described by Klindworth, et al. [23]. In the initial PCR step, the V3 to V4 region was amplified using specific primers with attached overhang adapters. In this study, the V3 to V4 regions were amplified using the 341F (5′– CCTACGGGNGGCWGCAG–3′) forward primer and the 805R (5′– GACTACHVGGGTATCTAATCC–3′) reverse primer which produced an expected amplicon size of ~460 bp. PCR reactions were performed in a total volume of 25 µL and included 12.5 µL 2× KAPA HiFi HotStart ReadyMix (KAPA Biosystems, Wilmington, MA, USA), 3 µL of both forward and reverse primers (10 µM each), 2.5 µL DNA template (50 ng µL^−1^) and 4 µL PCR grade water. A reaction mixture containing PCR-grade water (but no template DNA) was included as a negative control. The amplifications were performed in a Biometra TGradient thermocycler (Biometra GmbH, Göttingen, Germany) using the following conditions: 95 °C for 3 min for initial denaturation, 25 cycles of denaturation at 95 °C for 30 s, annealing at 55 °C for 30 s and extension at 72 °C for 30 s; the final extension was carried out at 72 °C for 5 min. Following amplification, 5 µL of the amplified product of each sample was resolved by electrophoresis in a 1.5% agarose gel containing 10% SYBR^®^ Safe DNA gel stain (Thermo Fisher Scientific, Mississauga, ON, Canada) and visualized using a ChemiDoc^TM^ XRS^+^ imager (Bio-rad Laboratories, Mississauga, ON, Canada). The remaining 20 µL of amplified product was then purified using Agencourt AMPure XP magnetic beads (Beckman Coulter Inc., Mississauga, ON, Canada) following the manufacturer’s instructions. Briefly, 20 µL of magnetic beads were added to the PCR product followed by two washing steps using 200 µL of 80% ethanol for 30 s. The ethanol was pipetted out and the magnetic beads were air-dried at room temperature for 10 min. A total of 52.5 µL of 10 mM Tris buffer (pH 8.5) was added to the beads and after centrifugation, 50 µL of clear supernatant was transferred into a sterile 0.2 mL PCR tube. An aliquot of the purified product (~5 µL amplicon^−1^) was subsequently evaluated by agarose gel electrophoresis as described above and the purified amplicons were stored at −20 °C until use.

#### 2.8.3. Index PCR, Purification, and Sequencing

A second PCR amplification step was performed to attach dual indexing primers and Illumina sequencing adapters to the purified PCR products. These PCR reaction mixtures consisted of 5 µL of the purified amplicon, 5 µL each of Illumina forward index primer (S5XX) and Illumina reverse index primer 1 (N7XX), 25 µL of 2× KAPA HiFi HotStart ReadyMix and 10 µL of PCR grade water. The PCR amplification steps were performed as described above; however; the number of cycles was reduced to 8. Following amplification, 5 µL of each PCR amplicon was separated on a 1.5% agarose gel and visualized to check for bands of the predicted size (~550 bp). For the final purification step, 56 µL of AMPure XP magnetic beads were added to the remaining PCR product, incubated for 2 min then washed twice with 200 µL of 80% ethanol. After the second washing step, the beads were air dried at room temperature for 5 min. The beads were then resuspended in 32.5 µL of 10 mM Tris buffer (pH 8.5) and a total volume of 30 µL of clear supernatant was transferred into a sterile 96-well PCR plate (Roche Diagnostics, Laval, QC, Canada). Purified amplicons were quantified using a Nanodrop spectrophotometer and normalized to a final concentration of 10 ng µL^−1^ using 10 mM Tris buffer (pH 8.5). The libraries were then submitted for sequencing at the University of Guelph’s Advanced Analysis Centre using the Illumina MiSeq system (San Diego, CA, USA) with 2 × 250 chemistry.

### 2.9. Bioinformatic Analysis

Once the sequencing was complete, the output sequence read files (fastq) were obtained from the BaseSpace^®^ Sequence Hub (Illumina cloud computing software). The 16S rRNA sequences were analyzed using Mothur version 1.42.3 [24] according to the MiSeq SOP [https://www.mothur.org/wiki/MiSeq_SOP], accessed 13 May 2021 [25]. Sequence reads which were smaller than 400 bp, larger than 500 bp, had more than eight consecutive bp, and were outside the V3/V4 region of the 16S rRNA were removed from the dataset. Filtered sequence reads were aligned to the SILVA reference database version 138.1 [26], pre-clustered to merge sequences with less than 2 bp difference, and chimeras were removed using the open-source tool VSEARCH [27]. Sequences were classified using the SILVA align reference and taxonomy files at a cut-off of 80% and taxon resembling chloroplasts, mitochondria, unknowns, archaea, and eukaryotes were removed. The dataset was too large to create a distance matrix, thus phylotype was used and then classified into operational taxonomic units (OTUs). Sequences were subsampled in order to normalize all samples to the lowest number of sequences per sample (i.e., 3838). Some samples failed PCR and sequencing, thus were removed (Table 1).

#### Statistical Analysis of Microbiota Data

Normal distribution and homogeneity of each dataset were determined using Shapiro-Wilk and Levene tests in RStudio version 1.3.1093 [28]. When needed, data were normalized by log, square-root, or arcsine transformation. All data are presented as means ± SE unless otherwise specified. For alpha-diversity, rarefaction.single and summary.single commands in Mothur were used on subsampled OTU datasets to generate rarefaction curves and diversity tables based on Good’s coverage, observed species, Shannon diversity (non-parametric), Chao-1 richness, and ACE diversity indices. Significant effects of each factor (e.g., diet and time) were determined using ANOVA, and differences between treatments were determined using Tukey HSD. For beta-diversity, Bray-Curtis distance matrixes were Square-root transformed and the significance of each factor (e.g., diet and time) was determined using Analysis of Similarity (ANOSIM) with the *adonis* function based on the vegan package in RStudio [29]. Beta diversity was plotted using 2D non-metric multidimensional scaling (NMDS). Linear discriminant analysis Effect Size (LefSe) was used to identify OTUs that explain differences between treatments using Kruskal-Wallis tests and a Linear Discriminant Analysis threshold of 2.0 [30]. A *p*-value below 0.05 was considered significant.

### 2.10. Enumeration of Splenic Bacterial Load

#### Splenic DNA Extraction and qPCR

Genomic DNA from spleen samples was extracted using the DNeasy Blood and Tissue^®^ extraction kit (Qiagen, Toronto, ON, Canada) according to the manufacturer’s instructions for the purification of total DNA from animal tissue using spin-columns. Approximately, 0.1 g of spleen tissue was used for each DNA extraction. The concentration and purity of the DNA sample were estimated twice using a Nanodrop spectrophotometer. A *rpoC* qPCR assay developed by Strepparava, Wahli, Segner, and Petrini [22] was performed as described by Jarau, Di Natale, Huber, MacInnes, and Lumsden [20]. All samples were tested in triplicate. A known positive biological control from a clinical case of *F. psychrophilum* infection was included in each experiment. A no DNA template control and a dilution series of *F. psychrophilum* FPG101 DNA were included in each test as standards in every experiment.

### 2.11. Statistical Analysis of Survival and Spleen Data

Survival curves and spleen data were calculated using GraphPad Prism 7.0 (GraphPad Software, Inc., San Diego, CA, USA). Survivorship curves were compared using the log-rank test. A two-way ANOVA was used, and if the overall two-way ANOVA had a *p*-value < 0.05, Tukey’s multiple comparisons was used as a post hoc test. The number of spleens that were *F. psychrophilum* positive was analyzed using a Chi-square test. The splenic bacterial load of different fish groups was compared using a one-way analysis of variance (ANOVA) and Tukey’s multiple-comparison test. A *p*-value of <0.05 was considered to be significant.

## 3. Results

### 3.1. Diversity of Rainbow Trout Intestinal Microbiota

DNA was extracted from 132 fecal samples collected during the trial. Following the 16S rRNA gene amplification and purification steps, most of the DNA templates from the FFN- and ERY-treated groups produced faint bands when evaluated by agarose gel and 119 amplicon libraries were found to be suitable for sequencing. The remaining 13 samples were not included due to the absence of a band of desired size or insufficient DNA concentration (<10 ng µL^−1^). 15,422,986 sequences were processed and reduced to 4,022,717 after bioinformatics analyses. 1.8% of the original sequences contained chloroplasts, mitochondria, unknowns, and eukaryotes. The dataset had a mean of 33,804 sequences per sample that was subsampled (normalized) to 3388 sequences per sample as shown in the rarefaction curve (Appendix A).

#### 3.1.1. Alpha Diversity

A significant effect of diet, time, and diet x time interaction (*p* < 0.01) was found for all four diversity indices: No. of OTUs, Shannon diversity, Chao1 richness, and ACE diversity (Table 2 and Table 3). Diversity indices, No. of OTUs, Shannon, and Chao1 in the intestine of Ca, E, and F treatments were significantly higher (*p* < 0.05) than Cb when all time points were pooled (Appendix A), although only F was higher than Cb at D24 (Appendix A). Except for Shannon, diversity indices between E and F treatments were not significantly different (*p* > 0.05). All alpha diversity indices were increased between D-11 and the later time points (*p* < 0.05), except Shannon at D24, indicating an effect of time (Figure 2). Within each group (diet × time), E and F treatments had significantly different (*p* < 0.05) No. of OTUs, Shannon, and Chao1 diversities whereas no differences were found for treatments Ca and Cb (Table 2 and Figure 3).

#### 3.1.2. Beta Diversity

Diet and time had significant (*p* < 0.001) effects on beta diversity based on PERMANOVA (Table 4). Within each group, beta diversity from at least one time point was different from D-11 p.i., while each time point for F had a significantly different beta diversity (*p* < 0.05). The NMDS plot on the Bray-Curtis distance matrix did not clearly distinguish clusters based on all four treatment groups sampled at all time points (Figure 4a), although there was clear separation at the D24 time point (Figure 4b).

At the phyla level, 25 taxa were identified and the top five with mean relative abundance > 1% were Proteobacteria (44%), Tenericutes (35%), Spirochaetota (11%), Desulfobacterota (6%), and Firmicutes (2%) (Figure 5). At the genus level, 484 taxa were found and the top nine with a relative abundance > 1% were *Mycoplasma* (35%), *Brevinema* (11%)*, Acinetobacter* (10%), *Aeromonas* (6%), *Pseudomonas* (5%), *Escherichia-Shigella* (5%), *Sphingomonas* (4%), *Allorhizobium-NPR* (3%), and *Deefgea* (2%) (Figure 6).

#### 3.1.3. Microbiota before and after Oral Antibiotic Treatment (-D11 and D0 p.i.)

At D-11 p.i., Proteobacteria was observed to be predominantly occupying the intestines of all fish groups followed by Tenericutes, Spirochaetota, Desulfobacterota, and Firmicutes. Meanwhile, at the genus level, *Mycoplasma* was most common followed by *Aeromonas* and *Brevinema*. At D0 p.i. (following prophylactic oral treatment), there was a shift in the composition of intestinal microbiota in the FFN-treated group (F). An increase in the population of Proteobacteria occurred while a decrease was noted in Tenericutes, Spirochaetota, and Desulfobacterota (Figure 5). At the genus level, a more diverse community was noted in group F with a high abundance of *Sphingomonas Escherichia-Shigella* and *Sphingomonadaceae* with a decrease in *Mycoplasma* (Figure 6).

#### 3.1.4. Microbiota after *F. psychrophilum* Infection (D12 & D24 p.i.)

Relative abundance of bacteria at the phyla and genera levels changed after *F. psychrophilum infection* on D12 and D24 p.i. (Figure 5 and Figure 6). In groups, F and E, Proteobacteria, specifically *Acinetobacter, Aeromonas,* and *Pseudomonas*, increased at D12 and then decreased at D24 with an increase in Tenericutes, specifically *Mycoplasma*. In groups Ca and Cb, there was a progressive increase in Tenericutes, specifically *Mycoplasma,* and a decrease in Proteobacteria, specifically *Brevinema*. At D24, the LefSe analysis of biomarkers found that *Brevinema*, *Deefgea,* and *Escherichia_Shigella* were associated with group Ca, whereas *Mycoplasma* was associated with group Cb (Figure 7). Furthermore, group F was associated with *Pseudomonas*, *Acinetobacter,* and eight others, whereas group E was associated with *Aeromonas* and *Crenobacter*.

### 3.2. Flavobacterium Psychrophilum Infection

Following infection, some of the infected fish exhibited early signs of *F. psychrophilum* infection (i.e., lethargy, inappetence, and darkening of skin color) before mortality started to occur. Fish in group Cb, followed by groups F and E, exhibited a marked reduction in appetite. The progress of the infection was described previously by Jarau, Di Natale, Huber, MacInnes, and Lumsden [20]. Splenic bacterial culture was carried out from infected fish and yellow-pigmented bacteria (YPB) colonies with a phenotype characteristic of *F. psychrophilum* were confirmed as *F. psychrophilum* using the 16S rRNA PCR amplification test of Orieux, Bourdineaud, Douet, Daniel and Le Henaff [19]. There was no detectable *F. psychrophilum* in the spleens of fish sampled on D11 and D0 p.i., nor during the acclimation period.

#### 3.2.1. Survival Curves

Of the three groups of fish infected with *F. psychrophilum*, group E had the highest cumulative survival (95%) followed by group F with a cumulative survival of 91% while group Cb had the lowest cumulative survival (90%) (Figure 8). Neither morbidity nor mortality was observed in the control group Ca throughout the infection trial. No significant difference was observed in the relative survival amongst any of the infected groups (F, E, and Cb), or between the infected groups and the mock-infected control (Ca).

#### 3.2.2. Splenic Bacterial Load

To determine the *F. psychrophilum* load after infection, a *rpoC* qPCR assay was performed on samples collected on D3, D6, D9, and D12 p.i. to determine the splenic bacterial load (mean ± SEM log_10_
*rpoC* copies reaction^−1^) (Figure 9). No *F. psychrophilum* was detected in the mock-infected, control group Ca. *Flavobacterium psychrophilum* was first detected on D3 p.i. and ranged from 1.61–1.80 log_10_
*rpoC* copies reaction^−1^ in the infected groups with no significant difference between infected groups. In group F, on D12 p.i., a significant increase (*p* > 0.05) in the splenic bacterial load was detected compared with D6 and D9.

## 4. Discussion

The first objective of this study was to determine the effects of prophylactic oral antibiotic treatment of florfenicol (F) and erythromycin (E) on the microbial communities present in rainbow trout intestines in uninfected and *F. psychrophilum*-infected fish. Samples collected on D-11 and D0 p.i. (before and after oral antibiotic treatment) and on D12 and D24 p.i. (after *F. psychrophilum* infection) were evaluated using culture-independent, next-generation sequencing (NGS) on the Illumina MiSeq platform.

Tenericutes, Proteobacteria, Desulfobacterota, and Firmicutes were the most abundant phyla with *Mycoplasma*, *Brevinema, Acinetobacter, Aeromonas,* and *Pseudomonas* the most abundant genera residing in the intestines of healthy and untreated rainbow trout juveniles (Figure 5 and Figure 6). Our findings are consistent with the previous NGS studies on rainbow trout [5,6,8,9,16,31,32,33], although the higher relative abundance of Firmicutes as well as lower abundances of Cyanobacteria, Bacteroidetes, and Actinobacteria were found in the previous studies. *Mycoplasma*, a genus within the Tenericutes phylum, has been commonly reported as the most dominant bacterial genus in the distal intestines of wild and farmed Atlantic salmon [34,35,36,37], as well as in a few studies on rainbow trout [7,8]. *Brevinema* is a genus in the Spirochaetota phylum and has only been found once in the intestine and gill of rainbow trout [38]. *Acinetobacter*, *Aeromonas,* and *Pseudomonas* are genera within the Proteobacteria phylum and have been commonly found at a low abundance (<10%) in the intestine of rainbow trout [5,7,8,9,32].

Results from this study show a rapid recovery in regards to alpha diversity in groups F and E after the fish were infected with *F. psychrophilum,* while the infected group Cb had a reduction in alpha diversity (Table 2, Figure 2 and Figure 3). Similarly, a study by Donati, Madsen, Middelboe, Strube, and Dalsgaard [17] infected rainbow trout fry with *F. psychrophilum* and found alpha diversity had high variation at 1 and 8 days p.i. and recovered at 33 days. These authors also found feeding florfenicol and infection with *F. psychrophilum* decreased the abundance of *Lactobacillus* and *Weissella* at 1-day p.i. as well as decreased *Carnobacteria* and *Vagococcus* at day 8. In contrast, the present study found no effects of antibiotic or infection on the genera above, which may be explained by differences in fish size (30–40 vs. 1–2 g), *F. psychrophilum* strains (FPG101 vs. FPS-S6) and rearing conditions (125 vs. 8 L tanks) between the two studies. In addition, we found that *Mycoplasma* and *Brevinema* mostly disappeared after antibiotic treatment while infected Cb fish had high abundance (Figure 6 and Figure 7). A study by Brown, Wiens, and Salinas [38] found that genetic lines of rainbow trout that were more susceptible to BCWD infection had a higher abundance of *Brevinema* and a lower abundance of *Mycoplasma* in their gills and intestines. Despite the uncommon observation of *Brevinema*, other studies have found that *Mycoplasma* is a common inhabitant in the gut at high abundances of up to 83% in rainbow trout [7,8] and 92% in Atlantic salmon [34,35,36,37]. *Mycoplasma* are characterized by small genomes and the absence of a cell wall, making them difficult to culture on media [39]. *Mycoplasma penetrans* has been highlighted as a candidate agent of transmissible tumors in the gut of zebrafish [40]. However, two recent studies found that the relative abundance of *Mycoplasma* was positively correlated with the body weight of Atlantic salmon [34,41]. Based on metagenomic analysis, Rasmussen, et al. [42] found that the functional potential of *Mycoplasma* was involved in the de novo synthesis of arginine and ammonia detoxification, which suggests a mutualistic relationship between *Mycoplasma* and their salmonid hosts. Despite the fact that *Mycoplasma* is often reported to be an obligate parasite, findings by Rasmussen, Villumsen, Duchêne, Puetz, Delmont, Sveier, Jørgensen, Præbel, Martin, Bojesen, Gilbert, Kristiansen, and Limborg [42] revealed they could be specifically adapted for ammonotelic hosts, such as most teleosts, due to the ability to utilize ammonia in the gut.

Fish administered with E and F had either similar or increased alpha diversity and altered beta diversity compared to the control (Ca) at 24 d (Table 2 and Table 3 and Figure 2). However, the significant effect of time makes it difficult to compare treatments and the increased diversity may be related to an increased number of potential pathogens, such as *Acinetobacter* and *Pseudomonas*, found in the F-treated fish. In addition, prophylactic treatment with E resulted in increased potential pathogens, such as *Aeromonas*, as well. Antibiotic treatment and vaccines are difficult since *Aeromonas* strains are known for their enhanced capacity to acquire and exchange antibiotic resistance genes in aquatic environments [43]. A study by Huyben, Chiasson, Lumsden, Pham, and Chowdhury [5] recently found that feeding a blend of organic acids and essential oils was able to reduce the abundance of *Acinetobacter* and *Aeromonas* in the intestine of rainbow trout. However, these genera contain many species of bacteria that are not pathogenic, thus more research is required to sequence bacteria to the species level.

The F-treated group had a significant shift in the Proteobacteria phylum and several genera whereas the E-treated group did not. A previous study has suggested that *Sphingomonas*, a member of phylum Proteobacteria, is able to hydrolyze F [44]. This may explain the abundance of this genus in the F-treated group. The ability of *Sphingomonas* to degrade F is consistent with the notion that it could be a potential source of (acquired) F-resistance in other pathogens and could affect the efficacy of F in treating BCWD outbreaks. An F-resistance gene was first detected from a fish pathogen, *Photobacterium damselae* ssp. *piscicida* [45], however, there is no evidence that this same gene is present in the *F. psychrophilum* isolate used. *F. psychrophilum* FPG101 was previously determined to be sensitive to both florfenicol and erythromycin [18].

The E-treated group had a significant increase in the abundance of *Methylobacterium-M*. It is possible that E reduced the relative abundance of the susceptible bacterial population and created a niche for greater proliferation by *Methylobacterium*. This bacterium is known to reside in various natural environments i.e., plants, soil, and freshwater aquatic environments [46,47]. Moreover, *Methylobacterium* has been identified as part of the skin microbiota in brook charr *Salvelinus fontinalis* [48] and intestinal microbiota in sea bass *Dicentrarchus labrax* [49].

The second objective was to investigate the susceptibility of florfenicol-, erythromycin-treated and untreated fish to *F. psychrophilum* infection. As expected, the cumulative survival rate of the untreated fish group was the lowest of all the groups, although not significant. However, the mortality was less than expected based on preliminary experiments with the same strain and infection dose where FPG101 produced 62% mortality [20]. It is possible that the fish were more resistant to *F. psychrophilum* experimental infection since the fish used in this study were obtained from a hatchery that was involved in a family-based selective breeding program to increase genetic resistance against BCWD. Prophylactic oral treatment of rainbow trout using F and E did not significantly (*p* > 0.05) affect the fish susceptibility (Figure 8) to *F. psychrophilum* infection, but further work with a more virulent strain or dose is required to rigorously compare these antibiotics. Groups of fish that received prophylactic treatment did have a lower splenic load of *F. psychrophilum* than infected fish without treatment (Cb) for the first 9 days (Figure 9).

There are several factors that influence the diversity of intestinal microbiota. The intestinal microbiota has been reported to be influenced by the type of fish-rearing system which may vary over time [50]. We noticed that although the fish were stocked in the same rearing system and held under the same conditions, there were differences in the intestinal microbiota composition of fish groups. For example, there was a difference in the diversity of intestinal microbiota between fish groups prior to antibiotic treatment (-D11 p.i.). The influence of tank surface and tank biofilm has been previously reported [51,52]. Lyons, Turnbull, Dawson, and Crumlish [51] reported that tank biofilm also influenced the intestinal microbiota composition in trout. Previous studies reported that the intestinal microbiota composition varies over time [53,54,55]. Similar to previous studies, we observed that the intestinal microbiota of all groups including the control group changed over time. Our findings reveal that prophylactic oral treatment with F and E did not significantly (*p* > 0.05) affect the susceptibility of treated fish to *F. psychrophilum* infection. Negative impacts of antibiotic treatment on fish intestinal microbiota have been previously reported as Carlson, et al. [56] reported that rifampicin changed the intestinal microbiota in mosquitofish which increased the fish susceptibility to disease and osmotic stress.

Although antimicrobial treatment can reduce or eliminate susceptible pathogens from bacterial communities, it may have a significant negative impact on the composition and functionality of the host’s microbiota. It has been demonstrated to contribute to some climactic dysbiosis and long-term shifts of intestinal microbiota, the emergence of antibiotic-resistant strains, as well as up-regulation of antibiotic-resistance genes (ARGs) [57,58]. In addition, some opportunistic pathogens may be able to occupy previously unavailable ecological niches due to less competition. Previous studies have shown that genetic material including antimicrobial resistance genes can be transferred from remaining normal endogenous populations to opportunistic or potentially pathogenic allochthonous bacteria in the intestinal tract [59]. Accordingly, antimicrobial agents should be used with caution. Future studies are needed to characterize the abundance of ARGs and intestinal microbial diversity in rainbow trout from different farms/locations to give greater insights into the interaction between ARGs and fish intestinal microbiota. More research is also needed to understand the role that alternatives to antimicrobial agents such as probiotics, prebiotics and chemical compounds have in the composition of the intestinal microbiota and in health and disease resistance of rainbow trout.

In conclusion, we demonstrated that both the prophylactic oral treatment of florfenicol and erythromycin as well as infection with *F. psychrophilum* altered the intestinal microbiota in rainbow trout juveniles. We found significant effects of the treatment group and time over 24 days post-infection on the alpha and beta diversity of intestinal microbiota, thus we recommend that future studies should account for natural changes in microbiota over time. Prophylactic antibiotic treatments with florfenicol increased the alpha diversity of bacteria in the intestine compared to erythromycin and both treatments changed the beta diversity with a lower abundance of *Mycoplasma* and *Brevinema* while *Acinetobacter* and *Aeromonas* increased, which are known pathogens. These results suggest a higher load of potential pathogens is present in the intestine after antibiotic treatment and bacterial infection that are not favorable to the host, further supporting the cautious use of antibiotics. This study demonstrates that prophylactic oral treatment with antibiotics florfenicol and erythromycin as well as *F. psychrophilum* infection changed the composition of intestinal microbiota in rainbow trout. It is well-known that orally administered antibiotics alter the gut bacterial composition in the short-term, while this study found the gut microbiome did not recover by day 24 p.i. and further long-term effects need to be investigated.

## Figures and Tables

**Figure 1 pathogens-12-00454-f001:**
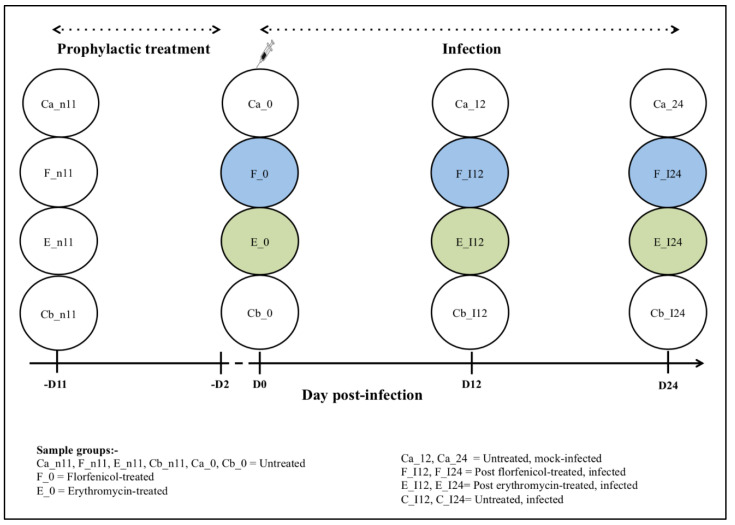
Experimental design indicating the periods of prophylactic treatment and infection. The intestinal contents from rainbow trout (n = 2–9 treatment^−1^) were collected at D-11 p.i., groups F and E were fed antibiotics until -D2 p.i., and then sampled at D0 p.i. before groups F, E, and Cb were infected with *F. psychrophilum* followed by sampling at D12 and D24 p.i. Group Ca was not fed antibiotics or infected (control).

**Figure 2 pathogens-12-00454-f002:**
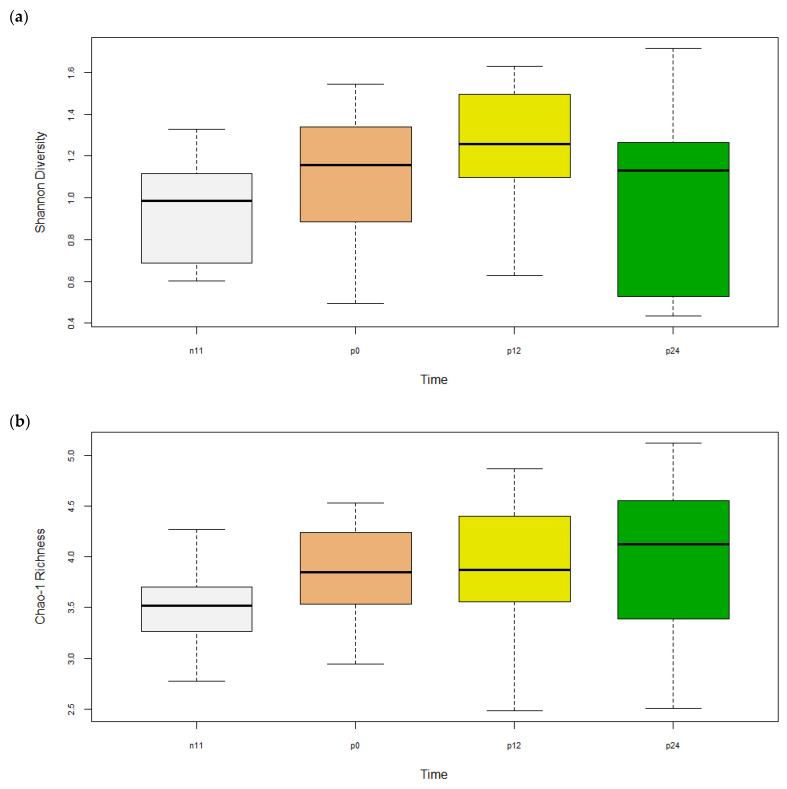
Alpha diversities of OTUs (mean ± SEM) in the intestine of rainbow trout pooled for each time point (n = 26–33 treatment^−1^) based on (**a**) Shannon Diversity, and (**b**) Chao-1 Richness (square-root transformed). Samples were collected at -D11, D0, D12, and D24 p.i. with *F. psychrophilum*. Differing letters indicate significant differences (*p* < 0.05).

**Figure 3 pathogens-12-00454-f003:**
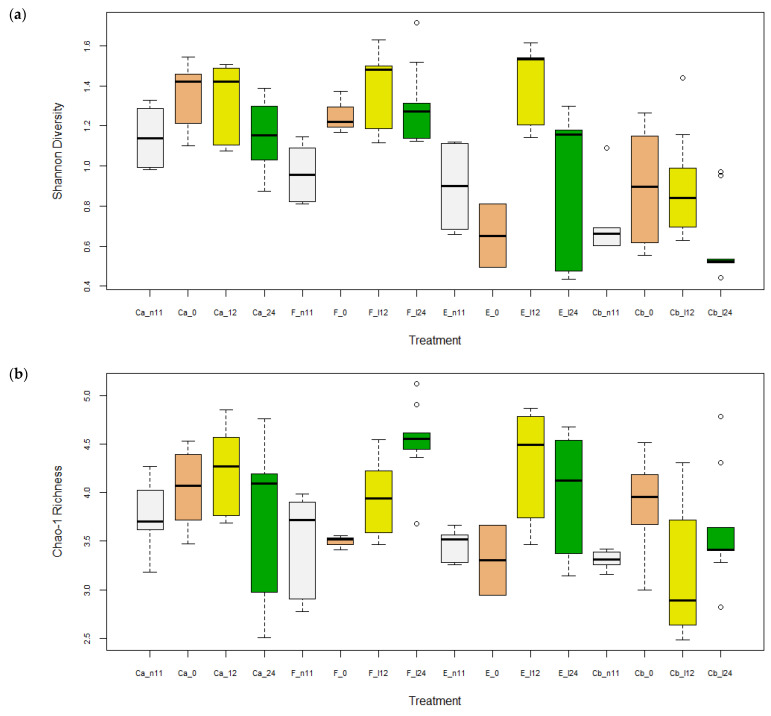
Alpha diversities of OTUs (mean ± SEM) in the intestine of rainbow trout for each diet (n = 2–9 treatment^−1^) based (**a**) Shannon Diversity, and (**b**) Chao-1 Richness (square root transformed). Treatments included control + uninfected (Ca), erythromycin + infected (E), florfenicol + infected (F), and control + infected (Cb) at days −11, 0, 12, and 24 post-infection with *F. psychrophilum*. Differing letters indicate significant differences (*p* < 0.05).

**Figure 4 pathogens-12-00454-f004:**
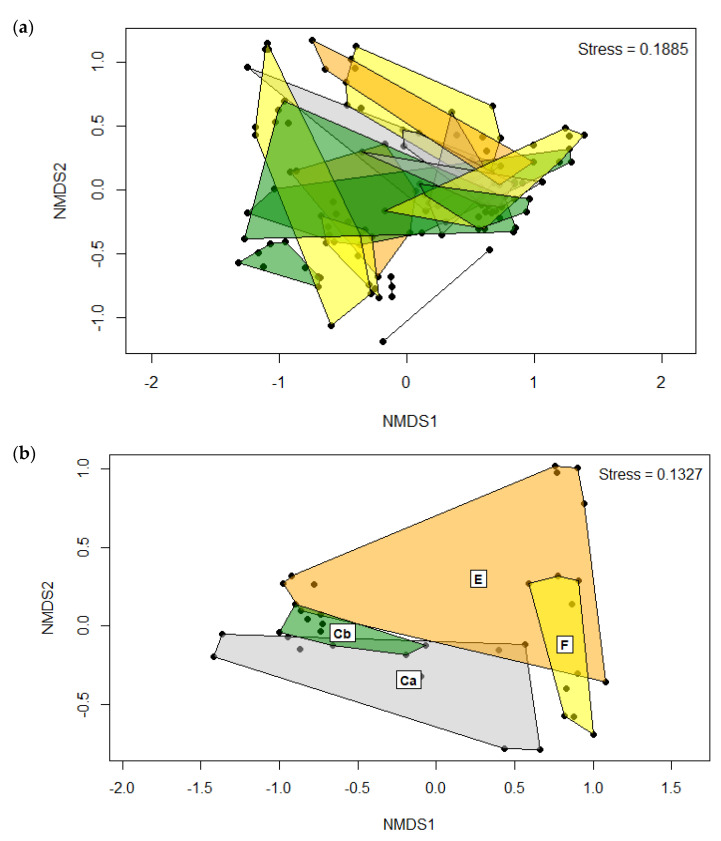
Beta diversity of OTUs in the intestine of rainbow trout (n = 2–9 treatment^_1^) at (**a**) all time points, and (**b**) D24 p.i. time point based on NMDS where control + uninfected (Ca), erythromycin + infected (E), florfenicol + infected (F), and control + infected (Cb).

**Figure 5 pathogens-12-00454-f005:**
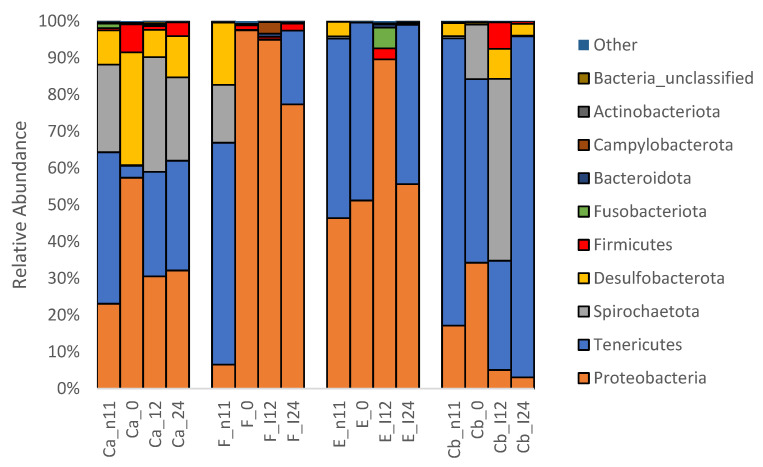
Mean relative abundance of OTUs in the intestine of rainbow trout (n = 2–9 treatment^−1^) at the phyla level on -D11, D0, D12, and D24 p.i. where control + uninfected (Ca), erythromycin + infected (E), florfenicol + infected (F), and control + infected (Cb).

**Figure 6 pathogens-12-00454-f006:**
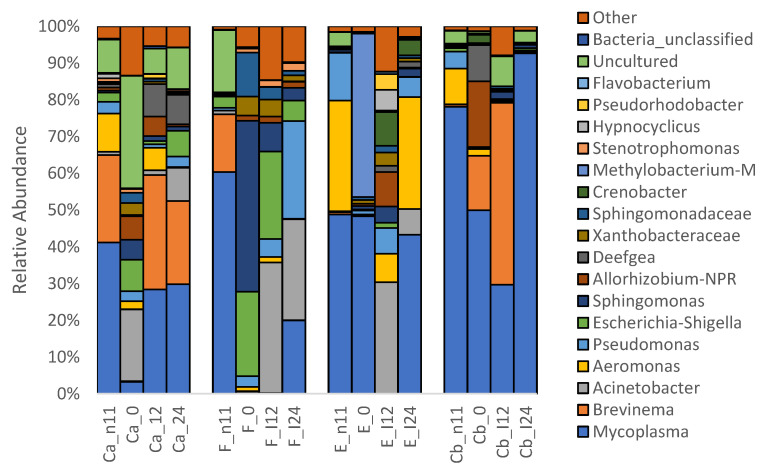
Mean relative abundance of OTUs in the intestine of rainbow trout (n = 2–9 treatment^−1^) at the genus level on -D11, D0, D12, and D24 p.i. where control + uninfected (Ca), erythromycin + infected (E), florfenicol + infected (F), and control + infected (Cb).

**Figure 7 pathogens-12-00454-f007:**
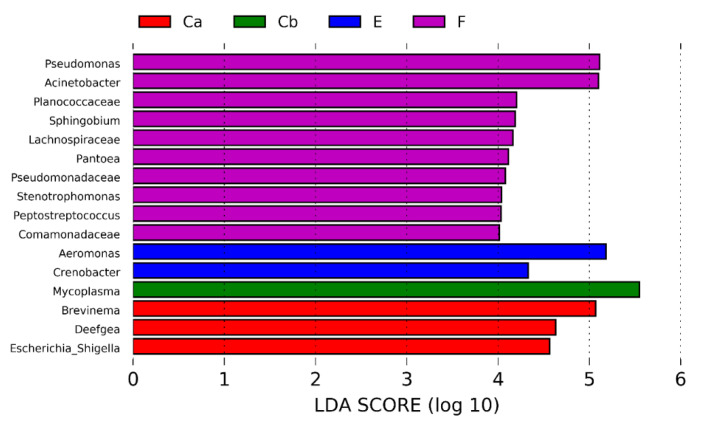
Biomarkers of OTUs in the intestine of rainbow trout (n = 2–9 treatment^−1^) at the genus level on D24 p.i. identified by LefSe where control + uninfected (Ca), erythromycin + infected (E), florfenicol + infected (F), and control + infected (Cb). All OTUs were significantly different (*p* < 0.05).

**Figure 8 pathogens-12-00454-f008:**
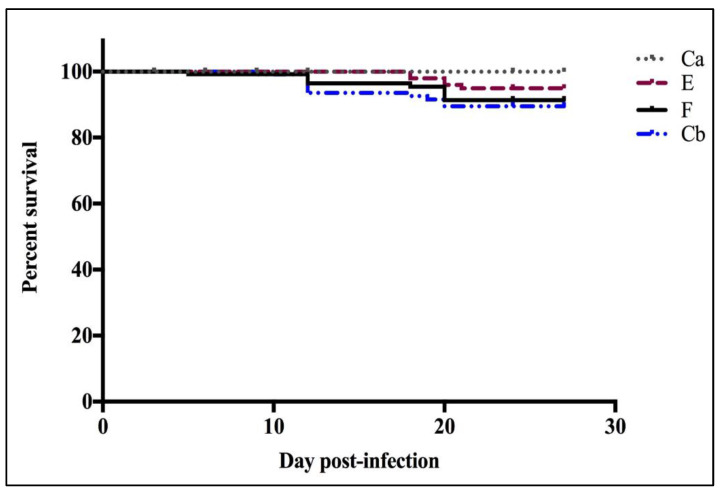
Cumulative survival curves of groups (n = 150 treatment^−1^) following i.p. infection with *F. psychrophilum* after prophylactic oral treatment with florfenicol (F), erythromycin (E), and untreated (Cb) rainbow trout were injected with FPG101 at 10^8^ cfu fish^−1^ compared to the triplicate mock-infected control group (Ca). Mortality was first observed on D6 p.i. There was no significant difference (*p* > 0.05) between the survival curves of all fish groups. Group E had the highest percent survival (95%) followed by group F (91%), and group Cb (90%). No mortalities occurred in the control fish group Ca.

**Figure 9 pathogens-12-00454-f009:**
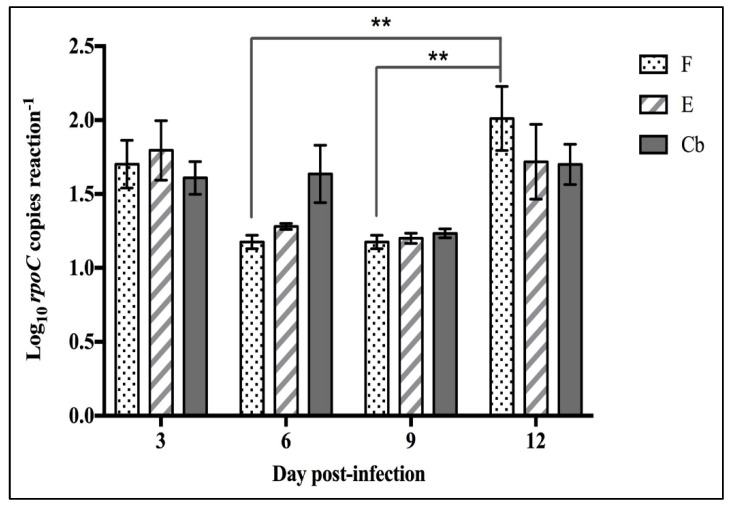
Splenic *F. psychrophilum* loads (mean ± SEM log_10_
*rpoC* copies reaction^−1^) of infected rainbow trout (n = 9 treatment^−1^) at four-time points (D3, D6, D9, and D12 p.i.) treated without antibiotics (Cb), with florfenicol (F) or with erythromycin (E). On D3 p.i., bacterial loads ranging from 1.61–1.80 log_10_
*rpoC* copies reaction^−1^ were detected in all groups. There was a significant increase (*p* < 0.05) in bacterial loads in the F group on D6 to D12 p.i., and D9 p.i. to D12 p.i. There was no significant change (*p* > 0.05) in splenic bacterial loads of fish groups E and Cb at any time. ** refers to statistically significant as *p* < 0.01.

**Table 1 pathogens-12-00454-t001:** Fish groups and day of sampling intestinal contents (n = 2–9 treatment^−1^). Samples groups at day 0 post-infection (p.i.) were collected after antibiotic treatment (groups F and E) and before the infection with *F. psychrophilum* FPG101 (groups F, E, and Cb).

Fish Group	Group Details	Intestinal Contents Sample Group	Day of Sampling (p.i.)	Sample Size
Ca	Control	Ca_n11 ^a^	−11	6
		Ca_0 ^a^	0	9
		Ca_12 ^b^	12	9
		Ca_24 ^b^	24	9
F	FFN-treated + FPG101-infected	F_n11 ^a^	−11	6
		F_0 ^d^	0	3
		F_I12	12	9
		F_I24	24	9
E	ERY-treated + FPG101-infected	E_n11 ^a^	−11	6
		E_0 ^e^	0	2
		E_I12	12	9
		E_I24	24	9
Cb	Untreated + FPG101-infected	Cb_n11 ^a^	−11	6
		Cb_0 ^c^	0	9
		Cb_I12	12	9
		Cb_I24	24	9

Note: Intestinal contents sample group is following the group details unless otherwise stated. ^a^ Untreated + uninfected. ^b^ Untreated + mock-infected. ^c^ Untreated, uninfected. ^d^ Florfenicol-treated + uninfected. ^e^ Erythromycin-treated + uninfected.

**Table 2 pathogens-12-00454-t002:** Alpha diversities of OTUs (mean ± SEM) in the intestine of rainbow trout (n = 2–9 treatment^−1^)_for each diet based on different indices.

Treatment	Control + Uninfected (Ca)				Erythromycin + Infected (E)				Florfenicol + Infected (F)				Untreated + Infected (Cb)				
Time (days)	−11	0	12	24	−11	0	12	24	−11	0	12	24	−11	0	12	24	SEM
Coverage	99.7	99.6	99.5	99.6	99.8	99.8	99.4	99.5	99.8	99.9	99.7	99.3	99.8	99.6	99.8	99.7	0.02
No. OTUs	31.5	50.6	48.3	37.0	24.5	25.0	58.6	43.3	23.5	32.7	44.8	64.4	22.3	30.1	22.6	26.4	1.9
Shannon	1.3	1.9	1.7	1.4	0.8	0.4	2.0	0.9	0.9	1.6	1.9	1.7	0.5	0.9	0.9	0.4	0.1
Chao1	44.9	62.5	72.3	51.9	32.4	29.0	81.2	61.8	36.7	33.0	55.2	99.0	27.4	52.3	28.5	44.6	3.0
Ace	46.4	67.5	84.0	73.0	38.7	28.5	97.6	75.5	46.1	33.6	64.0	128.1	28.8	74.0	35.6	61.9	4.1

SEM; standard error of the mean.

**Table 3 pathogens-12-00454-t003:** *p*-value table representing alpha diversity of OTUs in the intestine of rainbow trout (n = 2–9 treatment^−1^)_based on 2-way ANOVA that includes the effect of diet: control + uninfected (Ca), erythromycin + infected (E), florfenicol + infected (F) and control + infected (Cb) over four-time points: day −11, 0, 12, and 24 post-infection with *F. psychrophilum*.

Comparison			Coverage	No. OTUs	Shannon	Chao1	Ace
Diet:							
	Ca	Cb	0.188	<0.001	<0.001	0.004	0.048
	Ca	E	0.904	0.997	0.004	0.997	0.997
	Ca	F	0.994	0.947	0.999	0.976	0.986
	Cb	E	0.055	<0.001	<0.001	0.015	0.112
	Cb	F	0.356	<0.001	<0.001	0.002	0.028
	E	F	0.805	0.890	0.009	0.934	0.956
Time:							
	0	n11	0.379	0.003	0.016	0.039	0.038
	0	12	0.597	0.999	0.201	0.999	0.999
	0	24	0.065	0.973	0.087	0.992	0.654
	12	24	0.487	0.974	<0.001	0.973	0.663
	12	n11	0.015	0.001	<0.001	0.023	0.011
	24	n11	<0.001	0.004	0.778	0.007	<0.001
Diet × Time:							
Ca	0	n11	1.000	0.673	0.927	0.998	0.996
	0	12	0.990	1.000	1.000	1.000	1.000
	0	24	1.000	0.543	0.928	0.940	1.000
	12	24	0.999	0.753	0.996	0.656	0.995
	12	n11	0.652	0.839	0.994	0.944	0.804
	24	n11	1.000	1.000	1.000	1.000	1.000
Cb	0	n11	0.818	0.998	0.987	0.795	0.302
	0	12	0.655	0.729	1.000	0.222	0.309
	0	24	1.000	1.000	0.395	1.000	0.999
	12	24	0.982	0.988	0.339	0.837	0.956
	12	n11	1.000	1.000	0.978	1.000	1.000
	24	n11	0.995	1.000	1.000	0.998	0.920
E	0	n11	1.000	1.000	0.994	1.000	1.000
	0	12	0.416	0.479	0.003	0.518	0.438
	0	24	0.606	0.991	0.997	0.949	0.746
	12	24	1.000	0.885	<0.001	0.996	1.000
	12	n11	0.258	0.032	0.003	0.179	0.350
	24	n11	0.512	0.822	1.000	0.882	0.794
F	0	n11	0.799	0.997	0.896	1.000	1.000
	0	12	0.210	1.000	1.000	0.994	0.981
	0	24	<0.001	0.688	1.000	0.165	0.073
	12	24	0.115	0.940	1.000	0.525	0.368
	12	n11	1.000	0.230	0.047	0.950	0.995
	24	n11	0.017	0.002	0.270	0.018	0.029
Overall:							
	Diet		0.063	<0.001	<0.001	<0.001	0.017
	Time		<0.001	<0.001	<0.001	0.007	<0.001
	Diet × Time		<0.001	<0.001	0.002	<0.001	0.002

**Table 4 pathogens-12-00454-t004:** *p*-value table representing beta diversity of OTUs in the intestine of rainbow trout (n = 9 treatment^−1^) based on pairwise ANOSIM and PERMANOVA. Treatments included control + uninfected (Ca), erythromycin + infected (E), florfenicol + infected (F), and control + infected (Cb) over four-time points.

Group	Time (Days)		R-Value	*p*-Value
ANOSIM:				
Ca	0	n11	0.578	0.001
	0	12	0.692	<0.001
	0	24	0.490	<0.001
	12	24	0.012	0.345
	12	n11	0.055	0.232
	24	n11	0.065	0.722
Cb	0	n11	0.104	0.129
	0	12	0.224	0.009
	0	24	0.276	0.001
	12	24	0.581	<0.001
	12	n11	0.340	0.012
	24	n11	0.297	0.016
E	0	n11	0.604	0.070
	0	12	0.530	0.020
	0	24	0.231	0.060
	12	24	0.212	0.026
	12	n11	0.513	0.001
	24	n11	0.170	0.075
F	0	n11	0.963	0.011
	0	12	0.647	0.004
	0	24	0.776	0.001
	12	24	0.427	0.002
	12	n11	0.987	<0.001
	24	n11	0.638	<0.001
PERMANOVA:				
	Diet		0.176	0.001
	Time		0.110	0.001

## Data Availability

The data presented in this study are available on request from the corresponding author.

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
