# Peer review of "Impact of Infection with Flavobacterium psychrophilum and Antimicrobial Treatment on the Intestinal Microbiota of Rainbow Trout"

_pathogens, 2023, doi:10.3390/pathogens12030454_

Round 1

Reviewer 1 Report

The article focuses on the bacterial pathogen Flavobacterium psychrophilum, etiological agent of BCWD, and the impact of this bacterium and the antibiotics used for treatment (prophylactic) on the gut microbiome of rainbow trout juveniles. It also evaluates the susceptibility of fish that have received the antibiotics (prophylactically) to the infection of this bacterium. The study provides an important contribution in the fish gut microbiome area and the effects of disease and antimicrobial treatments. However, here some points that should be taken into considerations:

- Why prophylactic treatments? Is this a common practice or was this decided based on a scientific purpose? Please provide details on this choice.

- In general, the conclusions are not very strong. In addition, it is quite well-known that orally administered antibiotics alter the gut bacterial composition – a more interesting point is when and if the gut microbiome recover its original composition.

- A recent study looked into the gut microbiome of rainbow trout fry in relation to this bacterium and the antibiotic florfenicol before, during and after the infection (Donati et al. 2022. The Gut Microbiota of Healthy and Flavobacterium psychrophilum-Infected Rainbow Trout Fry Is Shaped by Antibiotics and Phage Therapies. Frontiers in Microbiology. DOI: 10.3389/fmicb.2022.771296). The authors should consider this work, since it is highly relevant and the topic is very similar.

- To follow from the previous point, I would try to be more specific e.g. in the title/abstract/conclusions – this new study focuses on the rainbow trout JUVENILES and on the INTESTINAL CONTENTS (so allochthonous communities).

- One point that is not taken into consideration is that studies focusing in the gut microbiome of rainbow trout distinguish between autochthonous and allochthonous communities. In this article authors focus on the whole intestinal content. This could be better highlighted.

- Figures and tables should be presented in numeric order: so tables 2 and 3 should be before table 4.

- Description of tables/figures must be improved. The number of sample (n) must be included and the description of the error bars (what do they represent).

- The pooling of data of various time points (per diet type) in the analysis and presentation of the results makes the results’ interpretation more complicated.

- It would be a very valuable addition to have a table (maybe in the SM) with all the metadata  for each sample that was collected (group, time, phenotypic data eg. fish weight or yes/no detection of bacterium)and/or successfully sequenced (e.g. yes/no sequenced, how many reads/samples).

ABSTRACT

Line 18: Clarify “Day”: When do you start counting (e.g. dpi – days post infection).

Line 20. Extra space between “genus.  Fish”

Line 26. Add juveniles after rainbow trout.

INTRODUCTION

Line 38. Rainbow trout is.

Line 46. I do not think research is lacking in relation to disease/antibiotics treatment. The focus have been the influence of diet but there are studies looking into this. Find references and re-phrase.

Line 50. BCWD and high mortalities. The infection with this bacterium can cause up to 90 % mortality in fry (RAINBOW TROUT FRY SYNDROME) and around 20% in juveniles and adult fish (BCWD). The mortality related to this infection highly depends on the size of the fish. Re-phrase and describe more specifically.

Line 52. Antimicrobial therapy (which are the antibiotics in use in Canada/regulations?)

Line 57-60. Not up to date.

Line 62-68. Please explain the prophylactic use of antibiotics – why?.

MATERIALS AND METHODS

Line 69. Add “and” between MATERIALS METHODS.

Line 73. At the beginning of a sentence, the number is written in letters. So “Fifty”.

Line 83-84. The method described by Orieux et al 2011: isn´t it qPCR?

Line 88. Please check English. “The medicated feed…was prepared…as described by”.

Line 99. Add info about the strain (e.g. serotype, when and from which organs the bacterium was isolated). After the reference, check English “isolated from a rainbow trout”.

Line 101, 127. As described by.

Line 113. Experimental design: there is no “control” fish that got the antibiotics and then no infection. So I find difficult to understand when the effect on the antibiotic alone on the gut bacteria is not there anymore (this is in the case they recover the original state). In order words it is not possible to say when and if the original gut composition is restored in relation to antibiotics alone.

Line 117. How big are the tanks? Add info. Where the tanks cleaned every day and the water replaced with new clean freshwater daily?

Line 120. Re-phrase, see here in bold: “Following the prophylactic treatment period, three groups [two treated groups (F & E) and an untreated group (Cb)] were injected with F. psychrophilum FPG101 whereas the untreated group Ca was mock-infected with CB and served as control”.

Line 124. Table 1. After title, I would add “samples at time 0 were collected before the infection”. I would also add explanation of “p.i.”. Maybe a new column could be added with the number of fish sampled per time point/per group.

Line 128, 129. Change “d” to “day/days”.

Line 133. Quite a high dose for IP injection. Why was such a dose chosen?

Line 139,140. Same as line 83-84. Is this the correct reference?

Line 142-148. Where other characteristics of the sampled fish recorded (e.g. fish weight, other analysis for bacterial/viral detection)? Were the intestinal contents and the spleens sampled from the same fish? I read that spleen samples were not collected for PCR (or qPCR) at –d11, d0 and d24, while instead these were the time points for the microbiome sampling. Is this correct? Do you have information about the infection status of the fish at d-11, d0 and d24? Please explain better.

Line 144. Add a full stop after p.i.. and change “d” to day (“One day prior to sampling…”).

Line 150. Describe the figure.

Line 158. Change “at different” to “at the selected”.

Line 159. Re-phrase: “the rpoC gene qPCR assay developed by”.

Line 164-172. Was a negative control for extraction included? This must be included to control for contamination.

Line 165. Is the selected DNA extraction kit optimized for DNA extraction for microbial profiling? Is there any previous work that have looked into its ability to break both Gram positive and negative bacteria? The protocol does not seem to include any mechanical disruption (bead beading), lysozyme treatment and proteinase K - these steps are generally included.

Line 173. Please check spelling “16S rRNA gene”.

Line 253. The “splenic DNA extraction” and “qPCR assay” could be under only one paragraph. The subparagraphs are not necessary. Check spelling.

Line 266. The title of the paragraph statistical analysis of spleen data is not correct as the paragraph starts with survival analysis. It should be changed.

RESULTS

Line 279. Which are these 13 samples that were excluded from the analysis?

Lines 283-285. These lines do not fit in the alpha diversity paragraph and should be moved up following the first part of the results section (after line 281).

Line 285. It seems quite a small amount of reads per samples (3,388 reads). How was the distribution of the reads per samples? It would be nice to see the rarefraction curves.

Line 287. Tables 2 and 3 are after table 4 and almost at the end of the article.

Line 289. It is not clear to me why pooling some of the data (the different time points per diet group). Some are infected and some not.

Line 289. Figure 2. Any of the fish was positive to the bacterium?

Line 306. Again I do not understand the pooling. In addition, some letters indicating significant differences are either outside the graph or missing.

Line 310. Again I do not find correct to pool all data per diet type together.

Line 311. “Within each group, beta diversity was different from D-11 p.i.,” to what? Is this correct?

Line 313. I find the pooling incorrect.

Line 316. Table 4. Is seems there is difference in the b-diversity between -11 at 0 in Ca but not in Cb (between the same time points), which should have been receiving the same diet at these particular time points. Why do you think? Do you have any phenotypic data to add? There is also significant difference in b-diversity in Ca between 0 and the other time points 12 and 24: may this be related to the fish growth or other phenotypic parameters? Any data about this? Differences in the b-diversity in Cb are observed between all time points except 0 vs -11: is this related to the infection? Any data about the recovery of the bacterium? It seems that treatment E has not effect (0 vs -11).

Line 324. Maybe it would interesting to see NMDS analysis divided/separated per time point. It may be more explanatory than pooling the data together. Have you tried other types of analysis, e.g. PCoA?

Line 328-350. I am not able to correlate the text to the figures as the legends of figure 5 and 6 are incomplete.

Line 359. The progress of the infection should be described.

Line 360. Which infected fish? Dead, moribund, sampled?

Line 363. Were all the “infected” fish positive to the bacterium? Please add info.

Line 372. It would have been highly relevant to see the infection load after 12 dpi since the last microbiome sample was taken 24 dpi.

Line 428. Table 2. Shannon diversity based on how many reads per samples?. SE=? The number of sample per experimental groups used for the analysis is missing. In addition, pooling the data in this manner does not show the variation of the values per time point in each of the feed group/with or without the infection.

Line 430-434. Figure 5 and 6. It is not possible to read and review these figures as the classification/legend is not complete. In addition, name of bacteria (phyla/genus) should be in one line. In figure 5, adjust x-axis labelling.

Line 455. Table 3. Is 2-way ANOVA the correct statistical test? I find this table confusing as it does not take into consideration the presence of infection (if present). In general I think, the pooling of data per diet type is not correct. Since there are many data/variables, it may be clearer if data would be presented per time point instead of being pooled.

Line 500. Maybe instead of writing which of the groups has the highest/lowest % survival, the actual numbers should be indicated (so 90, 91 and 95%).

Line 505. Remove extra space before “on 3D p.i.”.

DISCUSSION

Line 399. Remove extra space before Brevinema.

Line 405. This description of the second objective in the middle of the description of the first does not seem to fit. Maybe the sentence should be moved elsewhere (maybe 480?).

Line 406. Remove extra space before Results.

Line 445. I think it would be easier to compare the treatments among them if looked at one time points at a time.

Line 447,448. Increased potential pathogens? It should be better explain, these genera do not include only pathogenic bacteria.

 Line 480. There is no significant difference among the curves. So this cannot be said. Re-phrase.

Line 486. This is NOT true. Not SIGNIFICANT!

Line 486-490. This must be re-phrased.

Line 492-494. Are these differences significant compared to the control Cb? It does not seem like it. Re-phrase.

Line 544. Where precisely do you show the effect of the infection on the bacterial gut community? Unfortunately at this point, figures 5 and 6 are unreadable and the fact that data are pooled together and the infection status in not clearly explained makes these conclusions not very strong.

Line 553. Aeromonas and Pseudomonas are the most abundant genera in healthy and mock fish but in fish that have received antibiotics, their increased presence is related to increased load of potential pathogens. It should be explaind.

Line 543-558. I believe this last summarizing paragraph should be re-phrased after have better presented the results section. Table 4 clearly shows that the interpretation of the results is quite more intricate than antibiotics and infection change the status of the gut microbiome.  It seems there are other factors that affect the gut bacteria as also mentioned in the discussion (the b-diversity of control group Ca is different between time -11 and 0, something that does not appear in Cb). The article would benefits with a clearer presentation of the results (one time point at a time) and with more phenotypic data (e.g. something related to the fish growth as this has an impact on the fish gut microbiome, maybe water quality, presence of the bacterium).

Author Response

Letter from Co-authors – Response to Reviewers

To: Pathogens MDPI Journal

Manuscript ID: pathogens-1998414

Date: November 2022

Dear Editor,

Thank you to the editor, staff and reviewers for considering our manuscript for publication titled “Impact of infection with Flavobacterium psychrophilum and antimicrobial treatment on the intestinal microbiota of rainbow trout”. We know it takes a lot of time and effort to review and we really appreciate it. Below are the reviewers’ questions in bold and our responses below (not bold):

Reviewer 1

- Why prophylactic treatments? Is this a common practice or was this decided based on a scientific purpose? Please provide details on this choice.

We thank the reviewer for this valuable feedback. A sentence was added to the last paragraph in the introduction to explain this. ‘Florfenicol is often used to treat BCWD as soon as feed intake drops for a sustained period as waiting longer in an outbreak often leads to high mortality and multiple treatments. Strictly speaking this would be a prophylactic treatment and we were interested in the impact on subsequent infection that this use would have.’

- In general, the conclusions are not very strong. In addition, it is quite well-known that orally administered antibiotics alter the gut bacterial composition – a more interesting point is when and if the gut microbiome recover its original composition.

We thank the reviewer for this valuable feedback. We agree that antibiotic changes to the gut bacterial composition are well known, although effects of Flavobacterium psychrophilum have not been. Also, effects using advanced next-generation sequencing has rarely been performed to investigate these effects in rainbow trout. We also agree that the conclusions are not strong, so we have added “…juveniles that did not recover by day 24 p.i and included notable pathogens” to the abstract and conclusion.

- A recent study looked into the gut microbiome of rainbow trout fry in relation to this bacterium and the antibiotic florfenicol before, during and after the infection (Donati et al. 2022. The Gut Microbiota of Healthy and Flavobacterium psychrophilum-Infected Rainbow Trout Fry Is Shaped by Antibiotics and Phage Therapies. Frontiers in Microbiology. DOI: 10.3389/fmicb.2022.771296). The authors should consider this work, since it is highly relevant and the topic is very similar.

We thank the reviewer for this valuable feedback. This paper was published shortly before we submitted ours and we apologize for missing it. Donati et al. (2022) has been added to the references and we also added the following to the Discussion: “Similarly, a study by Donati et al. (2022) infected rainbow trout fry with F. psychrophi-lum and found alpha diversity decreased after 1 and 8 days p.i. and recovered after 33 days. These authors also found infection decreased abundance of Actinobacteria at 8 days p.i. and feeding florfenicol increased Actinobacteria after 1 day. In contrast, the present study found no effects on the abundance of Actinobacteria, as it was had <1% abundance, which may be explained by differences in fish size (30-40 vs 1-2 g).

- To follow from the previous point, I would try to be more specific e.g. in the title/abstract/conclusions – this new study focuses on the rainbow trout JUVENILES and on the INTESTINAL CONTENTS (so allochthonous communities).

We thank the reviewer for this valuable feedback. “(allochthonous bacteria)” was added to the abstract and methods. “Juveniles” is included in the abstract and conclusion and the size of the fish is in the abstract and methods as well.

- One point that is not taken into consideration is that studies focusing in the gut microbiome of rainbow trout distinguish between autochthonous and allochthonous communities. In this article authors focus on the whole intestinal content. This could be better highlighted.

We thank the reviewer for this valuable feedback. See above response.

- Figures and tables should be presented in numeric order: so tables 2 and 3 should be before table 4.

We thank the reviewer for this valuable feedback. Table 2 and 3 moved before table 4.

- Description of tables/figures must be improved. The number of sample (n) must be included and the description of the error bars (what do they represent).

We thank the reviewer for this valuable feedback. We have added “mean ± SEM” where applicable on all the tables and figures as well as number of samples (n).

- The pooling of data of various time points (per diet type) in the analysis and presentation of the results makes the results’ interpretation more complicated.

We thank the reviewer for this valuable feedback. We believe it is important to show both effects of diet and time because both had significant effects on alpha and beta diversity. It is difficult to assess microbiota changes due to diet if the microbiota is naturally changing over time as well. We added the following to the conclusion: “We found significant effects of treatment group and time over 24 days of infection on the alpha and beta diversity of intestinal microbiota, thus we recommend that future studies should account for natural changes in microbiota over time.”

- It would be a very valuable addition to have a table (maybe in the SM) with all the metadata  for each sample that was collected (group, time, phenotypic data eg. fish weight or yes/no detection of bacterium)and/or successfully sequenced (e.g. yes/no sequenced, how many reads/samples).

This is valid request however the authors feel that this is of little real value to the reader and lengthens a already quite long paper. This table was not included.

ABSTRACT

Line 18: Clarify “Day”: When do you start counting (e.g. dpi – days post infection).

We thank the reviewer for this valuable feedback. The “p.i.” was added throughout. Many text, tables and figures include day before the number, so this was kept separate for consistency.

Line 20. Extra space between “genus.  Fish”

We thank the reviewer for this valuable feedback. Revised as suggested.

Line 26. Add juveniles after rainbow trout.

We thank the reviewer for this valuable feedback. Revised as suggested.

INTRODUCTION

Line 38. Rainbow trout is.

We thank the reviewer for this valuable feedback. Revised as suggested.

Line 46. I do not think research is lacking in relation to disease/antibiotics treatment. The focus have been the influence of diet but there are studies looking into this. Find references and re-phrase.

We thank the reviewer for this valuable feedback. We agree there has been many studies that have looked at bacteria and disease, but there are not many that use next generation sequencing (NGS) over time. We rephrased this sentence: “Several studies have identified the bacteria present in the rainbow trout intestine and the influence of diet using NGS, although research is lacking on the effects of disease and antibiotic treatment, especially over time.”

Line 50. BCWD and high mortalities. The infection with this bacterium can cause up to 90 % mortality in fry (RAINBOW TROUT FRY SYNDROME) and around 20% in juveniles and adult fish (BCWD). The mortality related to this infection highly depends on the size of the fish. Re-phrase and describe more specifically.

We thank the reviewer for this valuable feedback. Very good point. We have rephrased these sentences to the following: “Rainbow trout are highly susceptible to bacterial cold-water disease (BCWD), also referred to as rainbow trout fry syndrome (RTFS), a disease caused by the Gram-negative filamentous bacterium Flavobacterium psychrophilum predominantly between 6-10 ËšC. RTFS and BCWD have been associated with high economic and production losses in the aquaculture industry since it has been reported to cause up to 90% mortality in fry and up to 20% mortality in juvenile and adult fish (Holt, 1987; Cipriano and Holt, 2005).”

Line 52. Antimicrobial therapy (which are the antibiotics in use in Canada/regulations?)

We thank the reviewer for this valuable feedback. We have added florfenicol and erythromycin.

Line 57-60. Not up to date.

We thank the reviewer for this valuable feedback. The outdated sentence about antibiotic studies on humans and mice has been deleted. We also added the following: “However, only one study has investigated the effects of florfenicol (Donati et al., 2022) and no studies have investigated the effects of erythromycin, especially on juvenile rainbow trout.”

We have also modified this paragraph to better explain the use of erythromycin. ‘No studies have investigated the effects of erythromycin, which has demonstrated efficacy for experimental BCWD (Jarau et al., 2019), but is only available on an emergency basis for fish in Canada.’

Line 62-68. Please explain the prophylactic use of antibiotics – why?.

We thank the reviewer for this valuable feedback. Please see first response above.

MATERIALS AND METHODS

Line 69. Add “and” between MATERIALS METHODS.

We thank the reviewer for this valuable feedback. Revised as suggested.

Line 73. At the beginning of a sentence, the number is written in letters. So “Fifty”.

We thank the reviewer for this valuable feedback. Revised as suggested.

Line 83-84. The method described by Orieux et al 2011: isn´t it qPCR?

We thank the reviewer for this valuable feedback. Revised as suggested.

Line 88. Please check English. “The medicated feed…was prepared…as described by”.

We thank the reviewer for this valuable feedback. Revised as suggested.

Line 99. Add info about the strain (e.g. serotype, when and from which organs the bacterium was isolated). After the reference, check English “isolated from a rainbow trout”.

We thank the reviewer for this valuable feedback. The following was added: “…was isolated in 2008 from the kidney of a rainbow trout…” This information is available in previous publications by Jarau as well.

Line 101, 127. As described by.

We thank the reviewer for this valuable feedback. Revised as suggested.

Line 113. Experimental design: there is no “control” fish that got the antibiotics and then no infection. So I find difficult to understand when the effect on the antibiotic alone on the gut bacteria is not there anymore (this is in the case they recover the original state). In order words it is not possible to say when and if the original gut composition is restored in relation to antibiotics alone.

We thank the reviewer for this valuable feedback. Yes, you are correct about the combined effect. We did consider this, but we were constrained by the number of tanks in the system (12 maximum) and this would add 6 more tanks. We will consider this in the design of future experiments.

Line 117. How big are the tanks? Add info. Where the tanks cleaned every day and the water replaced with new clean freshwater daily?

We thank the reviewer for this valuable feedback. The size of the tanks and water flow were mentioned in the “Fish and rearing conditions” section as follows: “…12 circular fiberglass tanks (125 L) with continuous aeration and flow-through ground water…”

Line 120. Re-phrase, see here in bold: “Following the prophylactic treatment period, three groups [two treated groups (F & E) and an untreated group (Cb)] were injected with F. psychrophilum FPG101 whereas the untreated group Ca was mock-infected with CB and served as control”.

We thank the reviewer for this valuable feedback. Revised as suggested.

Line 124. Table 1. After title, I would add “samples at time 0 were collected before the infection”. I would also add explanation of “p.i.”. Maybe a new column could be added with the number of fish sampled per time point/per group.

We thank the reviewer for this valuable feedback. The following was added: “Samples groups at day 0 post infection (p.i.) were collected after antibiotic treatment (groups F and E) and before the infection with F. psychrophilum FPG101 (groups F, E and Cb).”

Line 128, 129. Change “d” to “day/days”.

We thank the reviewer for this valuable feedback. Revised as suggested.

Line 133. Quite a high dose for IP injection. Why was such a dose chosen?

We thank the reviewer for this valuable feedback. Our previous study (Jarau et al., 2018) found this dose induced 51% mortality in rainbow trout. Lower doses only caused 17 and 22% mortality. This was added to the methods.

Line 139,140. Same as line 83-84. Is this the correct reference?

We thank the reviewer for this valuable feedback. Revised as suggested.

Line 142-148. Where other characteristics of the sampled fish recorded (e.g. fish weight, other analysis for bacterial/viral detection)? Were the intestinal contents and the spleens sampled from the same fish? I read that spleen samples were not collected for PCR (or qPCR) at –d11, d0 and d24, while instead these were the time points for the microbiome sampling. Is this correct? Do you have information about the infection status of the fish at d-11, d0 and d24? Please explain better.

We thank the reviewer for this valuable feedback. There is a sentence in ‘Fish and rearing conditions’ that describes the health screening performed. A sentence was modified in the results to explain that the five fish screened before experimentation did not have detectable F. psychrophilium. There was no detectable F. psychrophilum in the spleens of fish sampled on D11 and D0 p.i., and ‘nor during the acclimation period’ was added to clarify. Thus the population had no clinical signs, behaviour indications nor qPCR evidence for infection with or the presence of F. psychrophilum. Thus samping fish for qPCR before experimental infection was considered to be not necessary.

Intestinal content and spleens were not sampled from the same fish and since they were sampled at different time points as the reviewer has noted they could not be the same fish. Previous publications by Jarau et al. has illustrated why this was – after experimental infection fish are only qPCR positive for approx. 2 weeks. Therefore, we performed qPCR to confirm infection but did not extend the sampling.

Line 144. Add a full stop after p.i.. and change “d” to day (“One day prior to sampling…”).

We thank the reviewer for this valuable feedback. Revised as suggested.

Line 150. Describe the figure.

We thank the reviewer for this valuable feedback. The following was added: “The intestinal contents from rainbow trout (n = 9) were collected at day -11 p.i., groups F and E were fed antibiotics until day -2 p.i., them sampled at 0 p.i. before groups F, E and Cb were infected with F. psychrophilum followed by sampling at day 12 and 24 p.i.. Group Ca was not fed antibiotics or infected (control).”

Line 158. Change “at different” to “at the selected”.

We thank the reviewer for this valuable feedback. Revised as suggested.

Line 159. Re-phrase: “the rpoC gene qPCR assay developed by”.

We thank the reviewer for this valuable feedback. Revised as suggested.

Line 164-172. Was a negative control for extraction included? This must be included to control for contamination.

We thank the reviewer for this valuable feedback. We mentioned that sterile water was included as a negative control in the 16S rRNA gene amplification section. We ran the negative control on a 1% agarose gel and found no band, thus we assumed no contamination and did not sequence the sample. We do realize that some groups do sequence their blank samples to identify trace levels of contamination. We will perform this in future studies. Thank you for noting this.

Line 165. Is the selected DNA extraction kit optimized for DNA extraction for microbial profiling? Is there any previous work that have looked into its ability to break both Gram positive and negative bacteria? The protocol does not seem to include any mechanical disruption (bead beading), lysozyme treatment and proteinase K - these steps are generally included.

We thank the reviewer for this valuable feedback. The QIAamp DNA Stool Mini Kit has been used in several studies that analyzed intestinal content from fish (Huyben et al. 2020). Also, we mentioned in this section that we used heat-lysis at 95 ËšC for 5 min in order to lyse Gram positive bacteria. We did not bead beat, but we did vortex the samples and we included this in this section.

Line 173. Please check spelling “16S rRNA gene”.

We thank the reviewer for this valuable feedback. Revised as suggested.

Line 253. The “splenic DNA extraction” and “qPCR assay” could be under only one paragraph. The subparagraphs are not necessary. Check spelling.

We thank the reviewer for this valuable feedback. Revised as suggested.

Line 266. The title of the paragraph statistical analysis of spleen data is not correct as the paragraph starts with survival analysis. It should be changed.

We thank the reviewer for this valuable feedback. Revised as suggested.

RESULTS

Line 279. Which are these 13 samples that were excluded from the analysis?

This information is not presently available as the author Jarau has significant health issues and is not presently in communication.

Lines 283-285. These lines do not fit in the alpha diversity paragraph and should be moved up following the first part of the results section (after line 281).

We thank the reviewer for this valuable feedback. Revised as suggested.

Line 285. It seems quite a small amount of reads per samples (3,388 reads). How was the distribution of the reads per samples? It would be nice to see the rarefraction curves.

We thank the reviewer for this valuable feedback. We had a mean of 33,804 sequences per sample and that was subsampled (normalized) to 3,388 sequences per sample (we added this to the Results section). A study by Caporasoa et al (2011) PNAS found that 16S datasets subsampled to 2000 sequences per sample retained the same relationships compared to a dataset with 3 million sequences per sample. We included a rarefaction curve in the supplemental material.

Line 287. Tables 2 and 3 are after table 4 and almost at the end of the article.

We thank the reviewer for this valuable feedback. Revised as mentioned above.

Line 289. It is not clear to me why pooling some of the data (the different time points per diet group). Some are infected and some not.

We thank the reviewer for this valuable feedback. Please see comment above. Based on revealing effect of time.

Line 289. Figure 2. Any of the fish was positive to the bacterium?

We thank the reviewer for this valuable feedback. Yes, please see Figure 9.

Line 306. Again I do not understand the pooling. In addition, some letters indicating significant differences are either outside the graph or missing.

We thank the reviewer for this valuable feedback. Revised as suggested.

Line 310. Again I do not find correct to pool all data per diet type together.

We thank the reviewer for this valuable feedback. Beta diversity was not pooled per diet. Figure 4 shows all time points and then the 24 hour time point.

Line 311. “Within each group, beta diversity was different from D-11 p.i.,” to what? Is this correct?

We thank the reviewer for this valuable feedback. We add “from at least one time point…” to be more clear.

Line 313. I find the pooling incorrect.

We thank the reviewer for this valuable feedback. See same response for Line 310.

Line 316. Table 4. Is seems there is difference in the b-diversity between -11 at 0 in Ca but not in Cb (between the same time points), which should have been receiving the same diet at these particular time points. Why do you think? Do you have any phenotypic data to add? There is also significant difference in b-diversity in Ca between 0 and the other time points 12 and 24: may this be related to the fish growth or other phenotypic parameters? Any data about this? Differences in the b-diversity in Cb are observed between all time points except 0 vs -11: is this related to the infection? Any data about the recovery of the bacterium? It seems that treatment E has not effect (0 vs -11).

We thank the reviewer for this valuable feedback. We believe the intestinal microbiota can change significantly between sample points every 11-12 days when fed the same diet/treatment. We noted this in the Discussion and said the following: “Previous studies reported that the 517 intestinal microbiota composition varies over time (Ye et al., 2014; Narrowe et al., 2015; 518 Zarkasi et al., 2016). Similar to previous studies, we observed that the intestinal microbiota 519 of all groups including the control group, changed over time.”

Line 324. Maybe it would interesting to see NMDS analysis divided/separated per time point. It may be more explanatory than pooling the data together. Have you tried other types of analysis, e.g. PCoA?

We thank the reviewer for this valuable feedback. We believe the NMDS is the best analysis to plot the data since it does not rely on linear data and instead ranks samples against each other. We found that the day 24 p.i. NMDS plot was the clearest and most relevant to show the recovery of the intestinal microbiota communities. We could use many different plots, but we feel this indicates the separate clustering of the treatments and confirmed significant differences found in our results.

Line 328-350. I am not able to correlate the text to the figures as the legends of figure 5 and 6 are incomplete.

We thank the reviewer for this valuable feedback. Revised as suggested.

Line 359. The progress of the infection should be described.

We thank the reviewer for this valuable feedback. The reader can find more information in Jarau et al. (2018).

Line 360. Which infected fish? Dead, moribund, sampled?

We thank the reviewer for this valuable feedback. Yes, infected and then sampled as mentioned in the Methods section.

Line 363. Were all the “infected” fish positive to the bacterium? Please add info.

We thank the reviewer for this valuable feedback. Yes, please see Figure 9.

Line 372. It would have been highly relevant to see the infection load after 12 dpi since the last microbiome sample was taken 24 dpi.

We thank the reviewer for this valuable feedback. Yes, we predicted the fish would recover within 12 days, which was not the case. We will include a longer sampling point in future studies.

Line 428. Table 2. Shannon diversity based on how many reads per samples?. SE=? The number of sample per experimental groups used for the analysis is missing. In addition, pooling the data in this manner does not show the variation of the values per time point in each of the feed group/with or without the infection.

We thank the reviewer for this valuable feedback. Shannon diversity was based on the subsampled number of sequences per sample to reduce sequencing bias. The number of samples per group was added (n = 9) and SE was defined and added to the footer. The variation is included as error bars in the plots.

Line 430-434. Figure 5 and 6. It is not possible to read and review these figures as the classification/legend is not complete. In addition, name of bacteria (phyla/genus) should be in one line. In figure 5, adjust x-axis labelling.

We thank the reviewer for this valuable feedback. Revised as suggested.

Line 455. Table 3. Is 2-way ANOVA the correct statistical test? I find this table confusing as it does not take into consideration the presence of infection (if present). In general I think, the pooling of data per diet type is not correct. Since there are many data/variables, it may be clearer if data would be presented per time point instead of being pooled.

We thank the reviewer for this valuable feedback. Alpha diversities were log, square-root or arcsine-transformed if normal distribution and homogeneity assumptions were not met. Table 3 was revised to show p-values according to individual groups as suggested.

Line 500. Maybe instead of writing which of the groups has the highest/lowest % survival, the actual numbers should be indicated (so 90, 91 and 95%).

We thank the reviewer for this valuable feedback. Revised as suggested.

Line 505. Remove extra space before “on 3D p.i.”.

We thank the reviewer for this valuable feedback. Revised as suggested.

DISCUSSION

Line 399. Remove extra space before Brevinema.

We thank the reviewer for this valuable feedback. Revised as suggested.

Line 405. This description of the second objective in the middle of the description of the first does not seem to fit. Maybe the sentence should be moved elsewhere (maybe 480?).

We thank the reviewer for this valuable feedback. Revised as suggested.

Line 406. Remove extra space before Results.

We thank the reviewer for this valuable feedback. Revised as suggested.

Line 445. I think it would be easier to compare the treatments among them if looked at one time points at a time.

We thank the reviewer for this valuable feedback. Revised as suggested.

Line 447,448. Increased potential pathogens? It should be better explain, these genera do not include only pathogenic bacteria.

We thank the reviewer for this valuable feedback. We agree that you make a very good point so we have added the following: “However, these genera contain many species of bacteria that are not pathogenic, thus more research is required to sequence bacteria to the species level.”

Line 480. There is no significant difference among the curves. So this cannot be said. Re-phrase.

We thank the reviewer for this valuable feedback. We agree, so we added “not significant”.

Line 486. This is NOT true. Not SIGNIFICANT!

We thank the reviewer for this valuable feedback. We agree, so we have removed this sentence.

Line 486-490. This must be re-phrased.

We thank the reviewer for this valuable feedback. We agree, so we have removed this sentence.

Line 492-494. Are these differences significant compared to the control Cb? It does not seem like it. Re-phrase.

We thank the reviewer for this valuable feedback. Yes, we added the following: “than infected fish without treatment (Cb)”.

Line 544. Where precisely do you show the effect of the infection on the bacterial gut community? Unfortunately at this point, figures 5 and 6 are unreadable and the fact that data are pooled together and the infection status in not clearly explained makes these conclusions not very strong.

We thank the reviewer for this valuable feedback. Both the alpha and beta diversity were significantly affected in groups F, E and Cb. We have added non-pooled data for alpha diversity. We have revised Figures 5 and 6 as mentioned above. Figure 8 and 9 indicate the infection status as related to cumulative mortality and splenic load of F. psychrophilum. We believe our findings support these conclusions.

Line 553. Aeromonas and Pseudomonas are the most abundant genera in healthy and mock fish but in fish that have received antibiotics, their increased presence is related to increased load of potential pathogens. It should be explaind.

We thank the reviewer for this valuable feedback. We believe antibiotic treatment and F. psychrophilum infection modified the gut microbiota and created conditions that allowed pathogenic bacteria to proliferate, which are not favourable to the host. We have added the following: “and bacterial infection that are not favourable to the host,”

Line 543-558. I believe this last summarizing paragraph should be re-phrased after have better presented the results section. Table 4 clearly shows that the interpretation of the results is quite more intricate than antibiotics and infection change the status of the gut microbiome.  It seems there are other factors that affect the gut bacteria as also mentioned in the discussion (the b-diversity of control group Ca is different between time -11 and 0, something that does not appear in Cb). The article would benefits with a clearer presentation of the results (one time point at a time) and with more phenotypic data (e.g. something related to the fish growth as this has an impact on the fish gut microbiome, maybe water quality, presence of the bacterium).

We thank the reviewer for this valuable feedback. Yes, we have revised Figure 2, Table 2 (not pooled) and Table 4 as well as rephrased the conclusion. Thank you for your valuable input!

Reviewer 2 Report

The manuscript "Impact of infection with Flavobacterium psychrophilum and antimicrobial treatment on the intestinal microbiota of rainbow trout" describes changing of microbiota after antibiotic treatment. The topic seems quite interesting but i have a serious concern about its similarity index while checking plagiarism (around 68%). Please rewrite the manuscript carefully and submit it again.

Author Response

Reviewer 2

The topic seems quite interesting but I have a serious concern about its similarity index while checking plagiarism (around 68%). Please rewrite the manuscript carefully and submit it again.

We thank the reviewer for this valuable feedback. This study was part of a PhD thesis at the University of Guelph by Maureen Jarau, a coauthor on this study, that was published in 2018 and presumably this has been flagged by the plagiarism software. Prior to submission of the thesis it was checked using a plagiarism software and was not found to of concern. Pathogens accepts submissions that have previously been made available as preprints provided that they have not undergone peer review, which is true in this case.

Reviewer 3 Report

In the present work, Huyben and collaborators have analyzed the impact of two antimicrobial prophylactic treatments in the intestinal microbiota composition of rainbow trout, using 16S rRNA sequencing at different time points. Additionally, they also explored the effect of these treatments on the sensibility of fish to the infection of Flavobacterium psychrophilum. From their analysis, the authors concluded that the treatment with the two antimicrobial compounds (florfenicol and erythromycin) as well as F. psychrophilum infection have an impact in the gut microbiota composition in different ways.

 Although this manuscript is nicely written and contains interesting information for researchers working in rainbow trout aquaculture, especially in the field of microbiota, there are some comments or criticisms that must be solved:

1.     F. psychrophilum has been used as a pathogenic microorganism to evaluate the impact of the treatments on the infection outcome and microbiota composition. However, it is not justified the choice of this pathogen, mainly since this microorganism is not enteropathogenic and does not colonize the gut of infected fish.

2.     Concerning F. psychrophilum challenge, the growth protocol used for the bacterium is quite surprising. Why did the authors not grow the microorganism directly on CB instead of harvesting the colonies grown on CA and resuspending them on CB? With this protocol, you cannot manage the physiological state of bacteria. Furthermore, prior to the results included in the manuscript, a calibration challenge is strongly recommended to adjust the dose of infection, because changing the fish lines may affect the results, as happened in their study. Did the authors perform any pre-challenge with their pathogenic F. psychrophilum strain? Also, since the dose that they used for infection is high (compared to other studies using different F. psychrophilum strains), did the authors observe or identify any symptoms of infection in dead fish? In fact, detecting the pathogen in the spleen is not enough to state that an infection happened, mainly by molecular approaches. Did the authors were able to isolate the bacterium from any dead fish? For this last point, a good control will be to check if they are also able to detect F. psychrophilum RNA in non-dead infected fish to evaluate how reliable this method is as well as if the infection really happened or not in dead fish.

3.     The authors should deeply review the current literature about rainbow trout microbiota composition because they focus their discussion on a few papers (5-8 studies), and there is more work that has been done on this topic (i.e. https://doi.org/10.1128/AEM.00924-13; https://doi.org/10.1007/s11160-019-09558-y; or https://doi.org/10.2174/1874285801812010308). The same comment concerning microbiota changes and F. psychrophilum infection challenges (i.e. https://doi.org/10.1186/s42523-021-00107-2).

4.     The Discussion section could be improved since it is not well organized, and it is not easy to clearly follow it. The structure that they propose from the beginning is fine, presenting their two main objectives, and they should maintain it until the end. For the comparative analysis of intestinal microbiota composition, they only work with a “selected” (line 398) number of studies. The authors might also include other works where rainbow trout microbiota has been characterized. Also, a deeper analysis of the effects of F. psychrophilum infection on microbiota composition changes is missing: why this infection will induce changes at the intestinal microbiota composition level? why there is an increase of Mycoplasma and Brevinema genera? Of course, the authors cannot clearly answer to these questions from their data, but they can always hypothesize based on what has been described previously.

Minor comments:

-       Line 145: please, specify the benzocaine concentration used for fish euthanize.

-       Line 328: “top five” instead of “top four”, because you then mentioned 5 different taxonomic groups.

-       Line 384: to avoid repeating “rainbow trout”, I suggest replacing the second one by “fish”.

-       Lines 418-420: the sentence “Mycoplasma penetrans has been… (Burns et al., 2018)” does not contribute to the clarification of any point in this paragraph and I suggest removing it. Idem for the sentence on lines 452-454.

-       Lines 468-470: did the authors check if their F. psychrophilum FPG101 is sensible to florfenicol?

-       Lines 475-479: please, check the literature concerning to Methylobacterium genus since, to my knowledge, this microorganism has been detected in sea bass intestinal microbiota.

Author Response

Reviewer 3

  1. F. psychrophilum has been used as a pathogenic microorganism to evaluate the impact of the treatments on the infection outcome and microbiota composition. However, it is not justified the choice of this pathogen, mainly since this microorganism is not enteropathogenic and does not colonize the gut of infected fish.

We thank the reviewer for this valuable feedback. Actually, the use of F. psychrophilum is justified. Previous comments have clarified the need for a better explanation in the manuscript – see above – to allow readers to better appreciate this. Florfenicol is used when feeding rates go down over one or more days without diagnosis of the disease RTFS or BCWD because if diagnosis is performed treatment will be delayed. Delaying treatment means that mortality will be higher, more fish will go off feed – preventing treatment of the population as treatment is oral in feed, and multiple treatments of the same population will be needed. This is very expensive. So typical practice is to treat immediately when feeding rates go off and don’t recover within 1-2 days – each farm has its own protocol. Treatment is therefore by definition prophylactic. We also hypothesized that treatment would later the gut flora – one purpose of the study – and that this may make infection/mortality rates worse. We have seen subjective evidence for this in the field but not experimentally. F. psychrophilum is not an enteropathogenic organism but doesn’t need to be to justify the study. Hopefully the wording changes will explain this better.

  1. Concerning F. psychrophilum challenge, the growth protocol used for the bacterium is quite surprising. Why did the authors not grow the microorganism directly on CB instead of harvesting the colonies grown on CA and resuspending them on CB? With this protocol, you cannot manage the physiological state of bacteria. Furthermore, prior to the results included in the manuscript, a calibration challenge is strongly recommended to adjust the dose of infection, because changing the fish lines may affect the results, as happened in their study. Did the authors perform any pre-challenge with their pathogenic F. psychrophilum strain? Also, since the dose that they used for infection is high (compared to other studies using different F. psychrophilum strains), did the authors observe or identify any symptoms of infection in dead fish? In fact, detecting the pathogen in the spleen is not enough to state that an infection happened, mainly by molecular approaches. Did the authors were able to isolate the bacterium from any dead fish? For this last point, a good control will be to check if they are also able to detect F. psychrophilum RNA in non-dead infected fish to evaluate how reliable this method is as well as if the infection really happened or not in dead fish.

We thank the reviewer for this valuable feedback. Many of these questions are addressed in previous studies by Jarau et al. Growth on CA was used and is described. Suspension in CB was only for enumeration. This protocol is tried and tested in previous publications and at least one is referred to in the methods. F. psychrophilum forms a pellicle and is difficult to grow to sufficient numbers and to enumerate when grown only in CB. This protocol has been used successfully numerous times to infect fish and produce mortality, see Jarau publications and others. Preliminary trials are always performed to test virulence (this is mentioned in the discussion already) – this was the final effort in several theses that used F. psychrophilum to produce mortality in rainbow trout and for which preliminary trials were always used. This was not reported in detail in this case. This isolate was from the same bacterial stock stored frozen at -80C and used to infect trout and produce mortality as described in Jarau et al. 2019. The reason for the reduced virulence is likely fish size – and a breeding program to produce resistance in rainbow trout to BCWD – as mentioned in the discussion.

Symptoms of clinical disease were present – lethargy, colour changes, reduced feeding, mortality. This was also already described in the manuscript in the Results – F. psychrophilum infection – section. This same section also describes that the organism was cultured from the spleens of infected/affected fish.

  1. The authors should deeply review the current literature about rainbow trout microbiota composition because they focus their discussion on a few papers (5-8 studies), and there is more work that has been done on this topic (i.e. https://doi.org/10.1128/AEM.00924-13; https://doi.org/10.1007/s11160-019-09558-y; or https://doi.org/10.2174/1874285801812010308). The same comment concerning microbiota changes and F. psychrophilum infection challenges (i.e. https://doi.org/10.1186/s42523-021-00107-2).

We thank the reviewer for this valuable feedback. We have added a few of these studies to the Introduction and Discussion sections. We agree that they are relevant and should be cited, especially those related to NGS studies on rainbow trout.

  1. The Discussion section could be improved since it is not well organized, and it is not easy to clearly follow it. The structure that they propose from the beginning is fine, presenting their two main objectives, and they should maintain it until the end. For the comparative analysis of intestinal microbiota composition, they only work with a “selected” (line 398) number of studies. The authors might also include other works where rainbow trout microbiota has been characterized. Also, a deeper analysis of the effects of F. psychrophilum infection on microbiota composition changes is missing: why this infection will induce changes at the intestinal microbiota composition level? why there is an increase of Mycoplasma and Brevinema genera? Of course, the authors cannot clearly answer to these questions from their data, but they can always hypothesize based on what has been described previously.

We thank the reviewer for this valuable feedback. We have revised the Discussion as suggested and included more rainbow trout studies. As you said, we do not know why some bacteria are increasing and we do not want to add speculation. Thank you for you input.

Minor comments:

-       Line 145: please, specify the benzocaine concentration used for fish euthanize.

We thank the reviewer for this valuable feedback. We mentioned this in the first line of the Flavobacterium psychrophiluminfection section (50 mg/L).

-       Line 328: “top five” instead of “top four”, because you then mentioned 5 different taxonomic groups.

We thank the reviewer for this valuable feedback. Revised as suggested.

-       Line 384: to avoid repeating “rainbow trout”, I suggest replacing the second one by “fish”.

We thank the reviewer for this valuable feedback. Revised as suggested.

-       Lines 418-420: the sentence “Mycoplasma penetrans has been… (Burns et al., 2018)” does not contribute to the clarification of any point in this paragraph and I suggest removing it. Idem for the sentence on lines 452-454.

We thank the reviewer for this valuable feedback. We believe both references add to the context of other studies finding similar bacteria and their implications on fish health.

-       Lines 468-470: did the authors check if their F. psychrophilum FPG101 is sensible to florfenicol?

We thank the reviewer for this valuable feedback. This same isolate was used in a publication by Jarau et al. 2019 to determine sensitivity (relative to other isolates tested) to florfenicol and erythromycin. A sentence was added to indicate this in the discussion. ‘F. psychrophilum FPG101 was previously determined to be sensitive to both florfenicol and erythromycin (Jarau et al., 2019).’

-       Lines 475-479: please, check the literature concerning to Methylobacterium genus since, to my knowledge, this microorganism has been detected in sea bass intestinal microbiota.

We thank the reviewer for this valuable feedback. We have added the reference of this bacterium found in sea bass (Carda-Dieguez et al 2014).

Round 2

Reviewer 1 Report

I thank the authors for their work and appreciate their effort in answering my comments. However, I do not think the manuscript is ready for publication. I suggest the author to re-consider some of the already mentioned points (e.g. pooling of the data, inclusion of phenotypic data, explanation of non-inclusion of negative controls in relation to fish experiment and DNA extraction, a clearer visualization of the results and a more precise reporting of references). I highlight here in red some of my further comments that could be useful for your next submission.

General comments 

- In general, the conclusions are not very strong. In addition, it is quite well-known that orally administered antibiotics alter the gut bacterial composition – a more interesting point is when and if the gut microbiome recover its original composition.

 We thank the reviewer for this valuable feedback. We agree that antibiotic changes to the gut bacterial composition are well known, although effects of Flavobacterium psychrophilum have not been. Also, effects using advanced next-generation sequencing has rarely been performed to investigate these effects in rainbow trout. We also agree that the conclusions are not strong, so we have added “…juveniles that did not recover by day 24 p.i and included notable pathogens” to the abstract and conclusion.

I am not sure you are answering the question here. I would also be reluctant in writing ”and included notable pathogens”. I do not find this correct.

- A recent study looked into the gut microbiome of rainbow trout fry in relation to this bacterium and the antibiotic florfenicol before, during and after the infection (Donati et al. 2022. The Gut Microbiota of Healthy and Flavobacterium psychrophilum-Infected Rainbow Trout Fry Is Shaped by Antibiotics and Phage Therapies. Frontiers in Microbiology. DOI: 10.3389/fmicb.2022.771296). The authors should consider this work, since it is highly relevant and the topic is very similar.

 We thank the reviewer for this valuable feedback. This paper was published shortly before we submitted ours and we apologize for missing it. Donati et al. (2022) has been added to the references and we also added the following to the Discussion: “Similarly, a study by Donati et al. (2022) infected rainbow trout fry with F. psychrophi-lum and found alpha diversity decreased after 1 and 8 days p.i. and recovered after 33 days. These authors also found infection decreased abundance of Actinobacteria at 8 days p.i. and feeding florfenicol increased Actinobacteria after 1 day. In contrast, the present study found no effects on the abundance of Actinobacteria, as it was had <1% abundance, which may be explained by differences in fish size (30-40 vs 1-2 g).

I find this addition to the discussion incorrect i.e. the reported results obtained by Donati et al (2022) are not quite the ones described by the authors in the current manuscript.  E.g. I do not think this is correct: Similarly, a study by Donati et al. (2022) infected rainbow trout fry with F. psychrophi-lum and found alpha diversity decreased after 1 and 8 days p.i.. Alpha diversity does not seem to be decreased by the infection in this article 1 and 8 d.p.i., the authors report a high variation. Also: “and feeding florfenicol increased Actinobacteria after 1 day”. I do not see this in the article/do not understand your point ("after 1 day"). The reference should be added as being highly relevant but with the correct findings.

Please correct grammar (“, as it was had <1%).

 - It would be a very valuable addition to have a table (maybe in the SM) with all the metadata  for each sample that was collected (group, time, phenotypic data eg. fish weight or yes/no detection of bacterium)and/or successfully sequenced (e.g. yes/no sequenced, how many reads/samples).

 This is valid request however the authors feel that this is of little real value to the reader and lengthens a already quite long paper. This table was not included.

I think an overview table in the SM with all the metadata would help the reader to understand the dataset. 

introduction

Line 57-60. Not up to date.

 We thank the reviewer for this valuable feedback. The outdated sentence about antibiotic studies on humans and mice has been deleted. We also added the following: “However, only one study has investigated the effects of florfenicol (Donati et al., 2022) and no studies have investigated the effects of erythromycin, especially on juvenile rainbow trout.”

We have also modified this paragraph to better explain the use of erythromycin. ‘No studies have investigated the effects of erythromycin, which has demonstrated efficacy for experimental BCWD (Jarau et al., 2019), but is only available on an emergency basis for fish in Canada.’

 In line 65: Donati at al (2022) looked at effect of florfenicol and F. psychrophilum over time in rainbow trout fry. Please correct and include the information (missing info that I think are relevant to be included are underlined).

Materials and methods

Line 113. Experimental design: there is no “control” fish that got the antibiotics and then no infection. So I find difficult to understand when the effect on the antibiotic alone on the gut bacteria is not there anymore (this is in the case they recover the original state). In order words it is not possible to say when and if the original gut composition is restored in relation to antibiotics alone.

 We thank the reviewer for this valuable feedback. Yes, you are correct about the combined effect. We did consider this, but we were constrained by the number of tanks in the system (12 maximum) and this would add 6 more tanks. We will consider this in the design of future experiments.

This must be added to the manuscript. It is a very important point for the interpretation of the results.

Line 142-148. Where other characteristics of the sampled fish recorded (e.g. fish weight, other analysis for bacterial/viral detection)? Were the intestinal contents and the spleens sampled from the same fish? I read that spleen samples were not collected for PCR (or qPCR) at –d11, d0 and d24, while instead these were the time points for the microbiome sampling. Is this correct? Do you have information about the infection status of the fish at d-11, d0 and d24? Please explain better.

We thank the reviewer for this valuable feedback. There is a sentence in ‘Fish and rearing conditions’ that describes the health screening performed. A sentence was modified in the results to explain that the five fish screened before experimentation did not have detectable F. psychrophilium. There was no detectable F. psychrophilum in the spleens of fish sampled on D11 and D0 p.i., and ‘nor during the acclimation period’ was added to clarify. Thus the population had no clinical signs, behaviour indications nor qPCR evidence for infection with or the presence of F. psychrophilum. Thus samping fish for qPCR before experimental infection was considered to be not necessary.

Intestinal content and spleens were not sampled from the same fish and since they were sampled at different time points as the reviewer has noted they could not be the same fish. Previous publications by Jarau et al. has illustrated why this was – after experimental infection fish are only qPCR positive for approx. 2 weeks. Therefore, we performed qPCR to confirm infection but did not extend the sampling.

I see the point but I do not find this method scientifically strong and correct [indeed, the bacterial load in the spleen 12 dpi is quite high, similar to the ones measured 3 dpi]. This must also be added to the manuscript to inform the reader. It should be clear to the reader that spleen and intestine samples do not correspond to the same fish. Any phenotypic data? These are important points that need to be addressed in order to consolidate the findings observed in the NGS data. 

Line 150. Describe the figure.

 We thank the reviewer for this valuable feedback. The following was added: “The intestinal contents from rainbow trout (n = 9 treatment-1) were collected at day -11 p.i., groups F and E were fed antibiotics until day -2 p.i., them sampled at 0 p.i. before groups F, E and Cb were infected with F. psychrophilum followed by sampling at day 12 and 24 p.i.. Group Ca was not fed antibiotics or infected (control).”

 Check spelling.

Line 164-172. Was a negative control for extraction included? This must be included to control for contamination.

 We thank the reviewer for this valuable feedback. We mentioned that sterile water was included as a negative control in the 16S rRNA gene amplification section. We ran the negative control on a 1% agarose gel and found no band, thus we assumed no contamination and did not sequence the sample. We do realize that some groups do sequence their blank samples to identify trace levels of contamination. We will perform this in future studies. Thank you for noting this.

You ALSO need a negative control for the extraction procedure, not just the PCR amplification. Both procedures need a negative control to assure no contamination. This needs to be written.  

Author Response

Pathogens manuscript – round 2 responses (Jan 27 2023)

Responses for round 2 are in green.

Reviewer 1

I thank the authors for their work and appreciate their effort in answering my comments. However, I do not think the manuscript is ready for publication. I suggest the author to re-consider some of the already mentioned points (e.g. pooling of the data, inclusion of phenotypic data, explanation of non-inclusion of negative controls in relation to fish experiment and DNA extraction, a clearer visualization of the results and a more precise reporting of references). I highlight here in red some of my further comments that could be useful for your next submission.

General comments 

- In general, the conclusions are not very strong. In addition, it is quite well-known that orally administered antibiotics alter the gut bacterial composition – a more interesting point is when and if the gut microbiome recover its original composition.

 We thank the reviewer for this valuable feedback. We agree that antibiotic changes to the gut bacterial composition are well known, although effects of Flavobacterium psychrophilum have not been. Also, effects using advanced next-generation sequencing has rarely been performed to investigate these effects in rainbow trout. We also agree that the conclusions are not strong, so we have added “…juveniles that did not recover by day 24 p.i and included notable pathogens” to the abstract and conclusion.

I am not sure you are answering the question here. I would also be reluctant in writing ”and included notable pathogens”. I do not find this correct.

A: We thank the reviewer for this valuable feedback. We removed “and included notable pathogens” from the abstract and conclusion. We added “It is well-known that orally administered antibiotics alter the gut bacterial composition in the short-term, while this study found the gut microbiome did not recover by day 24 p.i. and further long-term effects need to be investigated”. We agree that the recovery time should be the focus.

- A recent study looked into the gut microbiome of rainbow trout fry in relation to this bacterium and the antibiotic florfenicol before, during and after the infection (Donati et al. 2022. The Gut Microbiota of Healthy and Flavobacterium psychrophilum-Infected Rainbow Trout Fry Is Shaped by Antibiotics and Phage Therapies. Frontiers in Microbiology. DOI: 10.3389/fmicb.2022.771296). The authors should consider this work, since it is highly relevant and the topic is very similar.

 We thank the reviewer for this valuable feedback. This paper was published shortly before we submitted ours and we apologize for missing it. Donati et al. (2022) has been added to the references and we also added the following to the Discussion: “Similarly, a study by Donati et al. (2022) infected rainbow trout fry with F. psychrophi-lum and found alpha diversity decreased after 1 and 8 days p.i. and recovered after 33 days. These authors also found infection decreased abundance of Actinobacteria at 8 days p.i. and feeding florfenicol increased Actinobacteria after 1 day. In contrast, the present study found no effects on the abundance of Actinobacteria, as it was had <1% abundance, which may be explained by differences in fish size (30-40 vs 1-2 g).

I find this addition to the discussion incorrect i.e. the reported results obtained by Donati et al (2022) are not quite the ones described by the authors in the current manuscript.  E.g. I do not think this is correct: Similarly, a study by Donati et al. (2022) infected rainbow trout fry with F. psychrophi-lum and found alpha diversity decreased after 1 and 8 days p.i.. Alpha diversity does not seem to be decreased by the infection in this article 1 and 8 d.p.i., the authors report a high variation. Also: “and feeding florfenicol increased Actinobacteria after 1 day”. I do not see this in the article/do not understand your point ("after 1 day"). The reference should be added as being highly relevant but with the correct findings.

A: We thank the reviewer for catching this. We replaced “decreased” with “high variation”. Also, the reviewer is correct that there was a significant increase in Actinobacteria infected with F. psychrophilum and fed phage treated feed, but not infection alone so we removed this sentence. We agree with the reviewer that the Donati et al. (2022) study should be compared to more in depth, so we added the following: “These authors also found feeding florfenicol and infection with F. psychrophilum decreased abundance of Lactobacillus and Weissella at 1 day p.i. as well as decreased Carnobacteria and Vagococcus at day 8. In contrast, the present study found no effects of antibiotic or infection on the genera above, which may be explained by differences in fish size (30-40 vs 1-2 g) and rearing conditions between the two studies.”

Please correct grammar (“, as it was had <1%).

A: We thank the reviewer for catching this. We removed this.

 - It would be a very valuable addition to have a table (maybe in the SM) with all the metadata  for each sample that was collected (group, time, phenotypic data eg. fish weight or yes/no detection of bacterium)and/or successfully sequenced (e.g. yes/no sequenced, how many reads/samples).

 This is valid request however the authors feel that this is of little real value to the reader and lengthens a already quite long paper. This table was not included.

I think an overview table in the SM with all the metadata would help the reader to understand the dataset. 

A: We thank the reviewer for this valuable feedback. Table 1 and Figure 1 already include data on group assignments (time) and experimental design. However, we added sample size to Table 1 and mentioned this in the Methods section. In the first paragraph of the results we also include the reads/sample after bioinformatic processing and total reads sequenced (3,388 and 15,422,986).

Introduction

Line 57-60. Not up to date.

 We thank the reviewer for this valuable feedback. The outdated sentence about antibiotic studies on humans and mice has been deleted. We also added the following: “However, only one study has investigated the effects of florfenicol (Donati et al., 2022) and no studies have investigated the effects of erythromycin, especially on juvenile rainbow trout.”

We have also modified this paragraph to better explain the use of erythromycin. ‘No studies have investigated the effects of erythromycin, which has demonstrated efficacy for experimental BCWD (Jarau et al., 2019), but is only available on an emergency basis for fish in Canada.’

 In line 65: Donati at al (2022) looked at effect of florfenicol and F. psychrophilum over time in rainbow trout fry. Please correct and include the information (missing info that I think are relevant to be included are underlined).

A: We thank the reviewer for catching this. We removed “juveniles” and added “and F. psychrophilum infection over time on rainbow trout fry”.

Materials and methods

Line 113. Experimental design: there is no “control” fish that got the antibiotics and then no infection. So I find difficult to understand when the effect on the antibiotic alone on the gut bacteria is not there anymore (this is in the case they recover the original state). In order words it is not possible to say when and if the original gut composition is restored in relation to antibiotics alone.

 We thank the reviewer for this valuable feedback. Yes, you are correct about the combined effect. We did consider this, but we were constrained by the number of tanks in the system (12 maximum) and this would add 6 more tanks. We will consider this in the design of future experiments.

This must be added to the manuscript. It is a very important point for the interpretation of the results.

A: We thank the reviewer for this valuable feedback. Good point. Under experimental design in the Methods section, we included the following: “Due to tank space limitations, we did not include a control group of fish that were not infected and administered antibiotics.”

Line 142-148. Where other characteristics of the sampled fish recorded (e.g. fish weight, other analysis for bacterial/viral detection)? Were the intestinal contents and the spleens sampled from the same fish? I read that spleen samples were not collected for PCR (or qPCR) at –d11, d0 and d24, while instead these were the time points for the microbiome sampling. Is this correct? Do you have information about the infection status of the fish at d-11, d0 and d24? Please explain better.

We thank the reviewer for this valuable feedback. There is a sentence in ‘Fish and rearing conditions’ that describes the health screening performed. A sentence was modified in the results to explain that the five fish screened before experimentation did not have detectable F. psychrophilium. There was no detectable F. psychrophilum in the spleens of fish sampled on D11 and D0 p.i., and ‘nor during the acclimation period’ was added to clarify. Thus the population had no clinical signs, behaviour indications nor qPCR evidence for infection with or the presence of F. psychrophilum. Thus samping fish for qPCR before experimental infection was considered to be not necessary.

Intestinal content and spleens were not sampled from the same fish and since they were sampled at different time points as the reviewer has noted they could not be the same fish. Previous publications by Jarau et al. has illustrated why this was – after experimental infection fish are only qPCR positive for approx. 2 weeks. Therefore, we performed qPCR to confirm infection but did not extend the sampling.

I see the point but I do not find this method scientifically strong and correct [indeed, the bacterial load in the spleen 12 dpi is quite high, similar to the ones measured 3 dpi]. This must also be added to the manuscript to inform the reader. It should be clear to the reader that spleen and intestine samples do not correspond to the same fish. Any phenotypic data? These are important points that need to be addressed in order to consolidate the findings observed in the NGS data. 

A: We thank the reviewer for this valuable feedback. We agree this can be clearer so we added the following to the spleen paragraph in the Methods section: “on p.i. day 3, 6, 9 and 12. These were not the same fish sampled for intestinal content and a maximum of 12 days was used based on previous detectable levels of F. psychrophilum by qPCR (Jarau et al., 2018).” Aside from initial weight, we do not have other phenotypic data to add since it was a short 24-day study and we apologize for this.

Line 150. Describe the figure.

 We thank the reviewer for this valuable feedback. The following was added: “The intestinal contents from rainbow trout (n = 9 treatment-1) were collected at day -11 p.i., groups F and E were fed antibiotics until day -2 p.i., them sampled at 0 p.i. before groups F, E and Cb were infected with F. psychrophilum followed by sampling at day 12 and 24 p.i.. Group Ca was not fed antibiotics or infected (control).”

 Check spelling.

A: We have revised the spelling throughout.

Line 164-172. Was a negative control for extraction included? This must be included to control for contamination.

 We thank the reviewer for this valuable feedback. We mentioned that sterile water was included as a negative control in the 16S rRNA gene amplification section. We ran the negative control on a 1% agarose gel and found no band, thus we assumed no contamination and did not sequence the sample. We do realize that some groups do sequence their blank samples to identify trace levels of contamination. We will perform this in future studies. Thank you for noting this.

You ALSO need a negative control for the extraction procedure, not just the PCR amplification. Both procedures need a negative control to assure no contamination. This needs to be written.  

A: We thank the reviewer for this valuable feedback. We included the following in the microbiota analysis of the Methods section: “A negative control sample was not included in DNA extraction, PCR amplification or sequencing and should be included in future studies to assure no contamination.”

Thank you reviewer 1 for your valuable contribution to our manuscript.

Reviewer 2 Report

no comments.

Author Response

Pathogens manuscript responses round 3 20230227

We thank the reviewers for their valuable feedback as they continue to improve the quality of our manuscript. We have included revisions to the manuscript and the responses are below in non-bolded font.

Reviewer #2

No further comments.

Reviewer 3 Report

In the current version of the manuscript, the authors correctly answer almost to all my comments, by adding the needed additional information when needed. I thank to the authors the fact that they clarify why and how the scientific question is justified. However, I believe that this is important to state from the Abstract to clearly highlight why their study is important.

Also, I still consider that the Discussion could be improved because it is still quite hard to follow. I recommend to the authors simplify it since there are many repetitions of the results that could be avoided.

Minor comments:

-       Lines 50-51: RTFS is not a “synonym” of BCWD, but is the disease that affects early development states of fish.

-       Figure 3 : the statistics in panel A are not clear.

-       Line 315:  “Chao” instead of “ChoA”.

-       Figures 5, 6, 7, 8 and 9 are not included in “Results” section but on the “Discussion”. Please, move them to the corresponding place.

-       Line 426: please, modify “select studies” by “a few studies” or similar.

-       Lines 434-440: are the infection routes the same between both studies? What about the F. psychrophilum strain used in Donati et al study?

-       Lines 484-485: even if I appreciate the fact that the authors took in consideration my comment about Aeromonas genus, I think that the idea about the pathogenic and non-pathogenic species within the genus Aeromonas is not well presented. I recommend to re-write this paragraph including needed bibliographic references concerning the Aeromonas genus.

-       Line 538: “significant” instead of “significance”.

-       Lines 575-593: this paragraph is quite redundant with the Results and Discussion section. Being the final conclusion, the authors should avoid to repeat again the results, but just give the final remarks and major conclusions of their work.

-       The new references (70 to 73) are not listed in alphabetic order.

Author Response

Pathogens manuscript – round 2 responses (Jan 27 2023)

Responses for round 2 are in green.

Reviewer 3

In the current version of the manuscript, the authors correctly answer almost to all my comments, by adding the needed additional information when needed. I thank to the authors the fact that they clarify why and how the scientific question is justified. However, I believe that this is important to state from the Abstract to clearly highlight why their study is important.

A: We thank the reviewer for this valuable feedback. We have revised the last concluding sentence in the Abstract and Discussion to be more clear.

Also, I still consider that the Discussion could be improved because it is still quite hard to follow. I recommend to the authors simplify it since there are many repetitions of the results that could be avoided.

A: We thank the reviewer for this valuable feedback. We agree that the results should not be repeated. We removed a few sentences to reduce repetition and simplify the Discussion.

Minor comments:

-       Lines 50-51: RTFS is not a “synonym” of BCWD, but is the disease that affects early development states of fish.

A: We thank the reviewer for this valuable feedback. Good point. We revised this to: “highly susceptible to rainbow trout fry syndrome (RTFS) and bacterial cold-water disease (BCWD) later on as juveniles and adults.”

-       Figure 3 : the statistics in panel A are not clear.

A: We adjusted the letters above each plot.

-       Line 315:  “Chao” instead of “ChoA”.

A: We revised this as suggested.

-       Figures 5, 6, 7, 8 and 9 are not included in “Results” section but on the “Discussion”. Please, move them to the corresponding place.

A: We cannot modify the layout, but we will let the copy editor know.

-       Line 426: please, modify “select studies” by “a few studies” or similar.

A: We revised this as suggested.

-       Lines 434-440: are the infection routes the same between both studies? What about the F. psychrophilum strain used in Donati et al study?

A: The route is the same (injected intraperitoneally), but they did use a different strain (FPS101 vs FPS-S6). I have added this potential difference between the two studies as well as fish size and rearing conditions.

-       Lines 484-485: even if I appreciate the fact that the authors took in consideration my comment about Aeromonas genus, I think that the idea about the pathogenic and non-pathogenic species within the genus Aeromonas is not well presented. I recommend to re-write this paragraph including needed bibliographic references concerning the Aeromonas genus.

A: This was not done. Information regarding the pathogenicity of Aeromonas sp. is well documented elsewhere. The reviewer has not requested expansion of this point for Acinetobacter and Pseudomonas. The authors strongly feel that this information is not required and doesn’t improve the manuscript.

-       Line 538: “significant” instead of “significance”.

A: We revised this as suggested.

-       Lines 575-593: this paragraph is quite redundant with the Results and Discussion section. Being the final conclusion, the authors should avoid to repeat again the results, but just give the final remarks and major conclusions of their work.

A: We removed two large sentences from the concluding paragraph to not repeat the results as the reviewer suggested.

-       The new references (70 to 73) are not listed in alphabetic order.

A: We revised this as suggested.

Thank you reviewer 3 for your valuable contribution to our manuscript.

Round 3

Reviewer 1 Report

I thank the authors for answering my comments. However, the manuscript is not ready for publication.

I find the data analysis quite problematic (i don´t find the data pooling the correct method to analyzing the dataset, indeed it seems there is an effect of time when data are pooled together but this is lost when looking singularly at each treamtment group) and also the data visualization is not clear (e.g.  table 2, 3 and figure 2, 3 are repetitive, unclear and not precisely described; as an example: the sample number for the data is indicated as 9 per treatment, however the correct numbers are only indicated in table 1 under "sample size" - this comment is valid for all your figures/tables).

other comments:

-line 53: is this the correct range of temperatures where it is possible to find the infection?

-alpha diversity values are very low (e.g. shannon and Chao1 index-table2) compared to other studies looking at fish gut microbiome. 

-please check y-axis of Chao1 in figures 2,3

-why two different qPCR methods were used?

-the references in the text are not formatted according to the numbering in the reference list.

Author Response

Pathogens manuscript responses round 3 20230227

We thank the reviewers for their valuable feedback as they continue to improve the quality of our manuscript. We have included revisions to the manuscript and the responses are below in non-bolded font.

Reviewer #1:

I find the data analysis quite problematic (i don´t find the data pooling the correct method to analyzing the dataset, indeed it seems there is an effect of time when data are pooled together but this is lost when looking singularly at each treatment group) and also the data visualization is not clear (e.g.  table 2, 3 and figure 2, 3 are repetitive, unclear and not precisely described; as an example: the sample number for the data is indicated as 9 per treatment, however the correct numbers are only indicated in table 1 under "sample size" - this comment is valid for all your figures/tables).

Thanks for the feedback. You are correct, so we have added “(n = 2-9 treatment-1)” to all the tables and figures. We have included both sets of individual and pooled data.

other comments:

-line 53: is this the correct range of temperatures where it is possible to find the infection?

Thanks for the feedback. Yes, in our experience it is.

-alpha diversity values are very low (e.g. shannon and Chao1 index-table2) compared to other studies looking at fish gut microbiome.

Thanks for the feedback. There were only 24-56 OTUs per treatment on average, so it makes sense this was lower than previous studies. Many factors can be at play here, such as effects of temperature, tank size, fish size, feed, etc.

-please check y-axis of Chao1 in figures 2,3

Thanks for the feedback. Chao1 was square-root transformed to normalize the data. This was added to both figures.

-why two different qPCR methods were used?

Thanks for the feedback. We have found that the rpoC qPCR assay generates better standard curves and range and is preferable when used to quantitate F. psychrophilum in tissues, including the spleen. We use the 16S assay for confirmation and reconfirmation for culture as it is/was a generally more widespread assay for F. psychrophilum identification. No changes were made to the manuscript.

-the references in the text are not formatted according to the numbering in the reference list.

Thanks for the feedback. Revised as suggested to match the guidelines for this journal. We also added author contributions, institutional review, consent and data availability statements.

Reviewer 3 Report

The authors answered and justified all my comments. I recommend their work for publication.

Just a minor comment:

- line 445: add " F. psychrophilum strains"

Author Response

Pathogens manuscript responses round 3 20230227

We thank the reviewers for their valuable feedback as they continue to improve the quality of our manuscript. We have included revisions to the manuscript and the responses are below in non-bolded font.

Reviewer #3

- line 445: add " F. psychrophilum strains"

Thanks for the feedback. Revised as suggested.